# Beltrami Flow and Neural Diffusion on Graphs

**Benjamin P. Chamberlain**[*]
Twitter Inc.
bchamberlain@twitter.com

**James Rowbottom**[*]
Twitter Inc.

**Davide Eynard**
Twitter Inc.

**Francesco Di Giovanni**
Twitter Inc.

**Xiaowen Dong**
University of Oxford

**Michael M. Bronstein**
Twitter Inc. and Imperial College London

## Abstract

We propose a novel class of graph neural networks based on the discretised *Beltrami flow*, a non-Euclidean diffusion PDE. In our model, node features are supplemented with positional encodings derived from the graph topology and jointly evolved by the Beltrami flow, producing simultaneously continuous feature learning and topology evolution. The resulting model generalises many popular graph neural networks and achieves state-of-the-art results on several benchmarks.

## 1 Introduction

The majority of graph neural networks (GNNs) are based on the message passing paradigm [30], wherein node features are learned by means of a non-linear propagation on the graph. Multiple recent works have pointed to the limitations of the message passing approach. These include; limited expressive power [80, 95, 9, 7], the related oversmoothing problem [60, 62] and the bottleneck phenomena [1, 93], which render such approaches inefficient, especially in deep GNNs. Multiple alternatives have been proposed, among which are higher-order methods [54, 7] and decoupling the propagation and input graphs by modifying the topology, often referred to as *graph rewiring*. Topological modifications can take different forms such as graph sampling [32], $k$NN [43], using the complete graph [86, 1], latent graph learning [89, 36], or multi-hop filters [92, 73]. However, there is no agreement in the literature on when and how to modify the graph, and a single principled framework for doing so.

A somewhat underappreciated fact is that GNNs are intimately related to diffusion equations [16], a connection that was exploited in the early work of Scarselli et al. [76]. Diffusion PDEs have been historically important in computer graphics [83, 11, 51, 64], computer vision [13, 18, 6], and image processing [65, 82, 90, 85, 26, 12], where they created an entire trend of variational and PDE-based approaches. In machine learning and data science, diffusion equations underpin such popular manifold learning methods as eigenmaps [5] and diffusion maps [20], as well as the family of PageRank algorithms [63, 14]. In deep learning, differential equations are used as models of neural networks [16, 19, 25, 94, 71, 98] and for physics-informed learning [72, 22, 75, 21, 81, 47].

**Main contributions** In this paper, we propose a novel class of GNNs based on the discretised non-Euclidean diffusion PDE in joint positional and feature space, inspired by the *Beltrami flow* [82] used two decades ago in the image processing literature for edge-preserving image denoising. We show that the discretisation of the spatial component of the Beltrami flow offers a principled view on positional encoding and graph rewiring, whereas the discretisation of the temporal component can replace GNN layers with more flexible adaptive numerical schemes. Based on this model, we introduce Beltrami Neural Diffusion (BLEND) that generalises a broad range of GNN architectures and shows state-of-the-art performance on many popular benchmarks. In a broader perspective, our

35th Conference on Neural Information Processing Systems (NeurIPS 2021).

approach explores new tools from PDEs and differential geometry that are less well known in the graph ML community.

## 2 Background

**Beltrami flow** Kimmel et al. [40, 82, 39] considered images as 2-manifolds (parametric surfaces) $(\Sigma, g)$ embedded in some larger ambient space as $\mathbf{z}(\mathbf{u}) = (\mathbf{u}, \alpha\mathbf{x}(\mathbf{u})) \subseteq \mathbb{R}^{d+2}$ where $\alpha \geq 0$ is a scaling factor, $\mathbf{u} = (u_1, u_2)$ are the 2D *positional coordinates* of the pixels, and $\mathbf{x}$ are the $d$-dimensional *colour* or *feature coordinates* (with $d = 1$ or 3 for grayscale or RGB images, or $d = k^2$ when using $k \times k$ patches as features [12]). In these works, the image is evolved along the gradient flow of a functional $S[\mathbf{z}, g]$ called the *Polyakov action [68]*, which roughly measures the smoothness of the embedding[1]. For images embedded in Euclidean space with the functional $S$ minimised with respect to *both* the embedding $\mathbf{z}$ and the metric $g$, one obtains the following PDE:

$$\frac{\partial \mathbf{z}(\mathbf{u}, t)}{\partial t} = \Delta_{\mathbf{G}} \mathbf{z}(\mathbf{u}, t), \qquad \mathbf{z}(\mathbf{u}, 0) = \mathbf{z}(\mathbf{u}), \quad t \geq 0, \tag{1}$$

and boundary conditions as appropriate. Here $\Delta_{\mathbf{G}}$ is the *Laplace-Beltrami operator*, the Laplacian operator induced on $\Sigma$ by the Euclidean space we embed the image into. Namely, the embedding of the manifold allows us to *pull-back* the Euclidean distance structure on the image: the distance between two nearby points $\mathbf{u}$ and $\mathbf{u} + \mathrm{d}\mathbf{u}$ is given by

$$\mathrm{d}\ell^2 = \mathrm{d}\mathbf{u}^\top \mathbf{G}(\mathbf{u})\mathrm{d}\mathbf{u} = \mathrm{d}u_1^2 + \mathrm{d}u_2^2 + \alpha^2 \sum_{i=1}^{d} \mathrm{d}x_i^2, \tag{2}$$

where $\mathbf{G} = \mathbf{I} + \alpha^2 (\nabla_{\mathbf{u}}\mathbf{x}(\mathbf{u}))^\top \nabla_{\mathbf{u}}\mathbf{x}(\mathbf{u})$ is a $2 \times 2$ matrix called the *Riemannian metric*. The fact that the distance is a combination of the positional component (distance between pixels in the plane, $\|\mathbf{u} - \mathbf{u}'\|$) and the colour component (distance between the colours of the pixels, $\|\mathbf{x}(\mathbf{u}) - \mathbf{x}(\mathbf{u}')\|$) is crucial as it allows edge-preserving image diffusion.

When dealing with images, the evolution of the first two components of $(z_1, z_2) = \mathbf{u}$ is a nuisance amounting to the reparametrisation of the manifold and can be ignored. For grayscale images (the case when $d = 1$ and $\mathbf{z} = (u_1, u_2, x)$), this is done by projection along the dimension $z_3$, in which case the Beltrami flow takes the form of an inhomogeneous diffusion equation of $x$,

$$\frac{\partial x(\mathbf{u}, t)}{\partial t} = \frac{1}{\sqrt{\det \mathbf{G}(\mathbf{u}, t)}} \mathrm{div} \left( \frac{\nabla x(\mathbf{u}, t)}{\sqrt{\det \mathbf{G}(\mathbf{u}, t)}} \right) \qquad t \geq 0. \tag{3}$$

The *diffusivity*

$$a = \frac{1}{\sqrt{\det \mathbf{G}}} = \frac{1}{\sqrt{1 + \alpha^2 \|\nabla x\|^2}} \tag{4}$$

determining the speed of diffusion at each point, can be interpreted as an *edge indicator*: diffusion is weak across edges where $\|\nabla x\| \gg 1$. The result is an adaptive diffusion [65] popular in image processing due to its ability to denoise images while preserving their edges. For cases with $d > 1$ (multiple colour channels), equation (3) is applied to each channel separately; however, the metric $\mathbf{G}$ couples the channels, which results in their gradients becoming aligned [38].

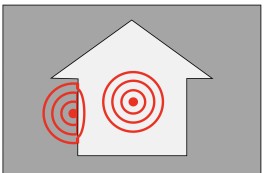

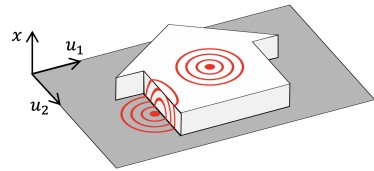

Figure 1: Two interpretations of the Beltrami flow: position-dependent bilateral kernel (top) and a Gaussian passed on the manifold (bottom).

**Special cases** In the limit case $\alpha = 0$, equation (3) becomes the simple homogeneous isotropic diffusion $\frac{\partial}{\partial t} x = \mathrm{div}(\nabla x) = \Delta x$, where $\Delta = \frac{\partial^2}{\partial u_1^2} + \frac{\partial^2}{\partial u_2^2}$ is the standard Euclidean Laplacian operator. The solution is

---

[1]Explicitly, by minimising the functional with respect to the embedding one finds Euler-Lagrange (EL) equations that can be used to dictate the evolution process of the embedding itself.

given in closed form as the convolution of the initial image and a Gaussian kernel with time-dependent variance,

$$x(\mathbf{u}, t) = x(\mathbf{u}, 0) \star \frac{1}{(4\pi t)^{d/2}} e^{-\|\mathbf{u}\|^2/4t} \tag{5}$$

and can be considered a simple linear low-pass filtering. In the limit $t \to \infty$, the image becomes constant and equal to the average colour.[2]

Another interpretation of the Beltrami flow is passing a Gaussian *on the manifold* (see Figure 1, bottom), which can locally be expressed as non-linear filtering with the *bilateral kernel* [85] dependent on the joint positional and colour distance (Figure 1, top),

$$x(\mathbf{u}, t) = \frac{1}{(4\pi t)^{d/2}} \int_{\mathbb{R}^2} x(\mathbf{v}, 0) e^{-\|\mathbf{u}-\mathbf{v}\|^2/4t} e^{-\alpha^2 \|\mathbf{x}(\mathbf{u},0)-\mathbf{x}(\mathbf{v},0)\|^2/4t} \mathrm{d}\mathbf{v}. \tag{6}$$

For $\alpha = 0$, the bilateral filter (6) reduces to a simple convolution with a time-dependent Gaussian.

## 3 Discrete Beltrami flow on graphs

We now develop the analogy of Beltrami flow for graphs. We consider a graph to be a *discretisation* of a continuous structure (manifold), and show that the evolution of the feature coordinates in time amounts to message passing layers in GNNs, whereas the evolution of the positional coordinates amounts to graph rewiring, which is used in some GNN architectures.

### 3.1 Graph Beltrami flow

Let $\mathcal{G} = (\mathcal{V} = \{1, \ldots, n\}, \mathcal{E})$ be an undirected graph, where $\mathcal{V}$ and $\mathcal{E}$ denote node and edge sets, respectively. We further assume node-wise $d$-dimensional features $\mathbf{x}_i \in \mathbb{R}^d$ for $i = 1, \ldots, n$. Denote by $\mathbf{z}_i = (\mathbf{u}_i, \alpha \mathbf{x}_i)$ the embedding of the graph in a joint space $\mathcal{C} \times \mathbb{R}^d$, where $\mathcal{C}$ is a $d'$-dimensional space with a metric $d_{\mathcal{C}}$ representing the node coordinates (for simplicity, we will assume $\mathcal{C} = \mathbb{R}^{d'}$ unless otherwise stated). We refer to $\mathbf{u}_i$ $\mathbf{x}_i$ and $\mathbf{z}_i$ as the *positional*, *feature* and *joint* coordinates of node $i$, respectively, and arrange them into the matrices $\mathbf{U}$, $\mathbf{X}$, and $\mathbf{Z}$, of sizes $n \times d'$, $n \times d$, and $n \times (d' + d)$.

For images, Beltrami flow amounts to evolving the embedding $\mathbf{z}$ along $\mathrm{div}(a(\mathbf{z})\nabla\mathbf{z})$, with $a$ a diffusivity map.[3] Thus, we consider the *graph Beltrami flow* to be the discrete diffusion equation

$$\frac{\partial \mathbf{z}_i(t)}{\partial t} = \sum_{j:(i,j)\in\mathcal{E}'} a(\mathbf{z}_i(t), \mathbf{z}_j(t))(\mathbf{z}_j(t) - \mathbf{z}_i(t)) \qquad \mathbf{z}_i(0) = \mathbf{z}_i; \quad i = 1, \ldots, n; \quad t \geq 0. \tag{7}$$

We motivate our definition as follows: $\mathbf{g}_{ij} = \mathbf{z}_j - \mathbf{z}_i$ and $\mathbf{d}_i = \sum_{j:(i,j)\in\mathcal{E}} \mathbf{g}_{ij}$ are the discrete analogies of the gradient $\nabla\mathbf{z}$ and divergence $\mathrm{div}(\mathbf{g})$, both with respect to a graph $(\mathcal{V}, \mathcal{E}')$ that can be interpreted as the numerical stencil for the discretisation of the continuous Laplace-Beltrami operator in (3). Note that $\mathcal{E}'$ can be different from the input $\mathcal{E}$ (referred to as 'rewiring'). As discussed in Section 3.3, most GNNs use $\mathcal{E}' = \mathcal{E}$ (input graph is used for diffusion, no rewiring). Alternatively, the positional coordinates of the nodes can be used to define a new graph topology either with $\mathcal{E}(\mathbf{U}) = \{(i, j) : d_{\mathcal{C}}(\mathbf{u}_i, \mathbf{u}_j) < r\}$, for some radius $r > 0$, or using $k$ nearest neighbours. This new rewiring is precomputed using the input positional coordinates (i.e., $\mathcal{E}' = \mathcal{E}(\mathbf{U}(0))$) or updated throughout the diffusion (i.e., $\mathcal{E}'(t) = \mathcal{E}(\mathbf{U}(t))$). Therefore, (7) can be compactly rewritten as

$$\frac{\partial \mathbf{z}_i(t)}{\partial t} = \mathrm{div}\left(a(\mathbf{z}(t))\nabla\mathbf{z}_i(t)\right).$$

The function $a$ is the diffusivity controlling the diffusion strength between nodes $i$ and $j$ and is assumed to be normalised: $\sum_{j:(i,j)\in\mathcal{E}'} a(\mathbf{z}_i, \mathbf{z}_j) = 1$. The dependence of the diffusivity on the

---

[2]Assuming appropriate boundary conditions.

[3]Note that in equation (3) the diffusivity function $a$ coincides with $(\det(\mathbf{G}(\mathbf{u}, t)))^{-\frac{1}{2}}$. Similarly to [82, Section 4.2] we have neglected the extra term $1/a$ appearing in (3).

embedding $\mathbf{z}$ matches the smooth PDE analysed in e.g. [82, Section 4.2] and is consistent with the form of attention mechanism used in e.g. [88, 86]. In matrix-form, we can also rewrite (7) as

$$\left(\tfrac{\partial}{\partial t}\mathbf{U}(t), \tfrac{\partial}{\partial t}\mathbf{X}(t)\right) = \left(\mathbf{A}(\mathbf{U}(t), \mathbf{X}(t)) - \mathbf{I}\right)\left(\mathbf{U}(t), \mathbf{X}(t)\right) \qquad (8)$$
$$\mathbf{U}(0) = \mathbf{U}; \ \mathbf{X}(0) = \alpha\mathbf{X}; \ t \geq 0,$$

where we emphasise the evolution of both the positional and feature components, coupled through the matrix-valued function $\mathbf{A}$

$$a_{ij}(t) = \begin{cases} a((\mathbf{u}_i(t), \mathbf{x}_i(t)), (\mathbf{u}_j(t), \mathbf{x}_j(t))) & (i,j) \in \mathcal{E}(\mathbf{U}(t)) \\ 0 & \text{else.} \end{cases}$$

representing the diffusivity. The graph Beltrami flow produces an evolution of the joint positional and feature coordinates, $\mathbf{Z}(t) = (\mathbf{U}(t), \mathbf{X}(t))$. In Section 3.3 we will show how the evolution of the feature coordinates $\mathbf{X}(t)$ results in feature diffusion or message passing on the graph, the core of GNNs. As noted in Section 2, in the smooth case the Beltrami flow is obtained as gradient flow of an energy functional when minimised with respect to *both* the embedding and the metric on the surface (an image). When the embedding takes values in the Euclidean space, this leads to equations of the form (3) with no channel-mixing and an exact form of the diffusivity determined by the pull-back $\mathbf{G}$ of the Euclidean metric. To further motivate our approach, it is tempting to investigate whether a similar conclusion can be attained here. Although in the discrete case the operation of pull-back is not well-defined, we are able to derive that the gradient flow of a *modified* graph Dirichlet energy gives rise to an equation of the form (7). We note though that the gradient flow does not recover the exact form of the diffusivity implemented in this paper. This is not a limitation of the theory and should be expected: by requiring the gradient flow to avoid channel-mixing and imitate the image analogy in [82] and by inducing a *discrete pull-back* condition, we are imposing constraints on the problem. We leave the theoretical implications for future work and refer to the Supplementary Materials for a more thorough discussion, including definitions and proofs.

**Theorem 1.** *Under structural assumptions on the diffusivity, graph Beltrami flow* (7) *is the gradient flow of the discrete Polyakov functional.*

## 3.2 Numerical solvers

**Explicit vs implicit schemes**   Equation (7) is solved numerically, which in the simplest case is done by replacing the continuous time derivative $\frac{\partial}{\partial t}$ with forward time difference:

$$\frac{\mathbf{z}_i^{(k+1)} - \mathbf{z}_i^{(k)}}{\tau} = \sum_{j:(i,j)\in\mathcal{E}(\mathbf{U}^{(k)})} a\left(\mathbf{z}_i^{(k)}, \mathbf{z}_j^{(k)}\right)\left(\mathbf{z}_j^{(k)} - \mathbf{z}_i^{(k)}\right). \qquad (9)$$

Here $k$ denotes the discrete time index (iteration) and $\tau$ is the time step (discretisation parameter). Rewriting (9) compactly in matrix-vector form with $\tau = 1$ leads to the *explicit Euler scheme*:

$$\mathbf{Z}^{(k+1)} = (\mathbf{A}^{(k)} - \mathbf{I})\mathbf{Z}^{(k)} = \mathbf{Q}^{(k)}\mathbf{Z}^{(k)}, \qquad (10)$$

where $a_{ij}^{(k)} = a(\mathbf{z}_i^{(k)}, \mathbf{z}_j^{(k)})$ and the matrix $\mathbf{Q}^{(k)}$ (diffusion operator) is given by

$$q_{ij}^{(k)} = \begin{cases} 1 - \tau \sum_{l:(i,l)\in\mathcal{E}} a_{il}^{(k)} & i = j \\ \tau a_{ij}^{(k)} & (i,j) \in \mathcal{E}(\mathbf{U}^{(k)}) \\ 0 & \text{otherwise} \end{cases}$$

The solution to the diffusion equation is computed by applying scheme (10) multiple times in sequence, starting from some initial $\mathbf{Z}^{(0)}$. It is 'explicit' because the update $\mathbf{Z}^{(k+1)}$ is done directly by the application of the diffusion operator $\mathbf{Q}^{(k)}$ on $\mathbf{Z}^{(k)}$ (as opposed to *implicit schemes* of the form $\mathbf{Z}^{(k)} = \mathbf{Q}^{(k)}\mathbf{Z}^{(k+1)}$ arising from backward time differences that require inversion of the diffusion operator [91]).

**Multi-step and adaptive schemes**   Higher-order approximation of temporal derivatives amount to using intermediate fractional steps, which are then linearly combined. Runge-Kutta (RK) [74, 44], ubiquitously used in numerical analysis, is a classical family of explicit numerical schemes, including Euler as a particular case. The Dormand-Prince (DOPRI) [24] is an RK method based on fifth and fourth-order approximations, the difference between which is used as an error estimate guiding the time step size [78].

**Adaptive spatial discretisation and rewiring**  Many numerical PDE solvers also employ adaptive spatial discretisation. The choice of the stencil (mesh) for spatial derivatives is done based on the character of the solution at these points; in the simulation of phenomena such as shock waves it is often desired to use denser sampling in the respective regions of the domain, which can change in time. A class of techniques for adaptive rewiring of the spatial derivatives are known as Moving Mesh (MM) methods [33]. Interpreting the graph $\mathcal{E}'$ in (7) as the numerical stencil for the discretisation of the continuous Laplace-Beltrami operator in (3), we can regard rewiring as a form of MM.

## 3.3  Relation to graph neural networks

Equation (9) has the structure of many GNN architectures of the 'attentional' type [10], where the discrete time index $k$ corresponds to a (convolutional or attentional) layer of the GNN and multiple diffusion iterations amount to a deep GNN. In the diffusion formalism, the time parameter $t$ acts as a continuous analogy of the layers, in the spirit of neural differential equations [19]. Typical GNNs amount to explicit single-step (Euler) discretisation schemes, whereas our continuous interpretation can exploit more efficient numerical schemes.

**GNNs as instances of graph Beltrami flow**  The graph Beltrami framework leads to a family of graph neural networks that generalise many popular architectures (see Table 1). For example, GAT [88] can be obtained as a particular setting of our framework where the input graph is fixed ($\mathcal{E}' = \mathcal{E}$) and only the feature coordinates $\mathbf{X}$ are evolved. Equation (10) in this case becomes

$$\mathbf{x}_i^{(k+1)} = \mathbf{x}_i^{(k)} + \tau \sum_{j:(i,j)\in\mathcal{E}} a\left(\mathbf{x}_i^{(k)}, \mathbf{x}_j^{(k)}\right)\left(\mathbf{x}_j^{(k)} - \mathbf{x}_i^{(k)}\right) \tag{11}$$

and corresponds to the update formula of GAT with a residual connection and the assumption of no non-linearity between the layers. The role of the diffusivity is played by a learnable parametric *attention* function, which is generally *time-dependent*: $a(\mathbf{z}_i^{(k)}, \mathbf{z}_j^{(k)}, k)$. This results in separate attention parameters per layer $k$, which can be learned independently. Our intentionally simplistic choice of a *time-independent* attention function amounts to *parameter sharing* across layers. We will show in Section 5.1 that this leads to a smaller model that is less likely to overfit.

Another popular architecture MoNet [57] uses linear diffusion of the features with weights dependent on the structure of the graph expressed as 'pseudo-coordinates', which can be cast as attention of the form $a(\mathbf{u}_i, \mathbf{u}_j)$. Transformers [87] can be interpreted as feature diffusion on a fixed complete graph with $\mathcal{E}' = \mathcal{V} \times \mathcal{V}$ [10]. Positional encoding (used in Transformers as well as in several recent GNN architectures [9, 27]) amounts to attention dependent on both $\mathbf{X}$ and $\mathbf{U}$, which allows the diffusion to adapt to the local structure of the graph; importantly, the positional coordinates $\mathbf{U}$ are *precomputed*. Similarly, DeepSets [97] and PointNet [70] architectures can be interpreted as GNNs applied on a graph with no edges ($\mathcal{E}' = \emptyset$), where each node is treated independently of the others [10]. DIGL [43] performs graph rewiring as a pre-processing step using personalised page rank as positional coordinates, which are then fixed and not evolved. In the point cloud methods, DGCNN [89] and DGM [36], the graph is constructed based on the *feature coordinates* $\mathbf{X}$ and rewired adaptively (in our notation, $\mathcal{E}' = \mathcal{E}(\mathbf{X}(t))$).

| Method | Evolution | Diffusivity | Graph $(\mathcal{V}, \mathcal{E}')$ | Discretisation |
|---|---|---|---|---|
| ChebNet | Features $\mathbf{X}$ | Fixed $a_{ij}$ | Fixed $\mathcal{E}$ | Explicit fixed step |
| GAT | Features $\mathbf{X}$ | $a(\mathbf{x}_i, \mathbf{x}_j)$ | Fixed $\mathcal{E}$ | Explicit fixed step |
| MoNet | Features $\mathbf{X}$ | $a(\mathbf{u}_i, \mathbf{u}_j)$ | Fixed $\mathcal{E}$ | Explicit fixed step |
| Transformer | Features $\mathbf{X}$ | $a((\mathbf{u}_i, \mathbf{x}_i), (\mathbf{u}_j, \mathbf{x}_j))$ | Fixed $\mathcal{E} = \mathcal{V} \times \mathcal{V}$ | Explicit fixed step |
| DeepSet/PointNet | Features $\mathbf{X}$ | $a(\mathbf{x}_i)$ | Fixed $\mathcal{E} = \emptyset$ | Explicit fixed step |
| DIGL* | Features $\mathbf{X}$ | $a(\mathbf{x}_i, \mathbf{x}_j)$ | Fixed $\mathcal{E}(\mathbf{U})$ | Explicit fixed step |
| DGCNN/DGM* | Features $\mathbf{X}$ | $a(\mathbf{x}_i, \mathbf{x}_j)$ | Adaptive $\mathcal{E}(\mathbf{X})$ | Explicit fixed step |
| **Beltrami** | Positions $\mathbf{U}$ +Features $\mathbf{X}$ | $a((\mathbf{u}_i, \mathbf{x}_i), (\mathbf{u}_j, \mathbf{x}_j))$ | Adaptive $\mathcal{E}(\mathbf{U})$ | Explicit adaptive step / Implicit |

Table 1: GNN architectures interpreted as particular instances of our framework. *Attentional variant.

**Graph rewiring**  Multiple authors have recently argued in favor of decoupling the input graph from the graph used for diffusion. Such rewiring can take the form of graph sampling [32] to address scalability issues, data denoising [43], removal of information bottlenecks [1], or larger multi-hop filters [73]. The graph construction can also be made differentiable and a task-specific rewiring can

be learned [89, 36]. The statement of Klicpera et al. [43] that 'diffusion improves graph learning', leading to the eponymous paradigm (DIGL), can be understood as a form of diffusion on the graph connectivity independent of the features. Specifically, the authors used as node positional encoding the Personalised PageRank (PPR), which can be interpreted as the steady-state of a diffusion process

$$\mathbf{U}_{\text{PPR}} = \sum_{k \geq 0} (1 - \beta)\beta^k \mathbf{\Delta}_{\text{RW}}^k = (1 - \beta)(\mathbf{I} - \beta\mathbf{\Delta}_{\text{RW}})^{-1}, \quad 0 < \beta < 1, \tag{12}$$

where $\mathbf{\Delta}_{\text{RW}}$ is the random walk graph Laplacian and $\beta \in (0, 1)$ is a parameter such that $1 - \beta$ represents the restart probability. The resulting positional encoding of dimension $d = n$ can be used to rewire the graph by $k$NN sampling, which corresponds to using $\mathcal{E}' = \mathcal{E}(\mathbf{U}_{\text{PPR}})$ in our framework.

**Numerical schemes**  All the aforementioned GNN architectures can be seen as an explicit discretisation of equation (7) with fixed step size. On the other hand, our continuous diffusion framework offers an additional advantage of employing more efficient numerical schemes with adaptive step size. Graph rewiring of the form $\mathcal{E}' = \mathcal{E}(\mathbf{U}(t))$ can be interpreted as adaptive spatial discretisation (moving mesh method).

### 3.4  Extensions

**Non-Euclidean geometry**  There are multiple theoretical and empirical arguments [52, 17] in favor of using hyperbolic spaces to represent real-life 'small-world' graphs (in particular, scale-free networks can be obtained as $k$NN graphs in such spaces [8]). Our framework allows using a non-Euclidean metric $d_{\mathcal{C}}$ for the positional coordinates $\mathbf{U}$ (Figure 2). In Section 5 we show that hyperbolic positional encodings allow for a significant reduction in model size with only a marginal degradation of performance.

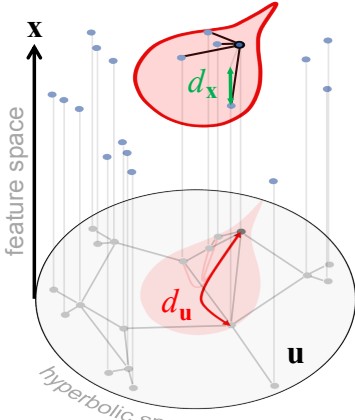

Figure 2: Graph Beltrami flow with hyperbolic positional coordinates.

**Time-dependent diffusivity**  The diffusivity function $a$ which we assumed time-independent and which lead to parameter sharing across layers (i.e., updates of the form $\mathbf{Z}^{(k+1)} = \mathbf{Q}(\mathbf{Z}^{(k)}, \boldsymbol{\theta})\mathbf{Z}^{(k)}$) can be made time-dependent of the form $\mathbf{Z}^{(k+1)} = \mathbf{Q}(\mathbf{Z}^{(k)}, \boldsymbol{\theta}^{(k)})\mathbf{Z}^{(k)}$, where $\boldsymbol{\theta}$ and $\boldsymbol{\theta}^{(k)}$ denote shared and layer-dependent parameters, respectively.

**Onsager diffusion**  As we noted, the Beltrami flow diffuses each channel separately. A more general variant of diffusion allowing for feature mixing is the *Onsager diffusion* [61] of the form $\frac{\partial}{\partial t}\mathbf{Z}(t) = \mathbf{Q}(\mathbf{Z}(t))\mathbf{Z}(t)\mathbf{W}(t)$, where the matrix-valued function $\mathbf{W}$ acts across the channels. GCN [42] can be regarded a particular setting thereof, with update of the form $\mathbf{X}^{(k+1)} = \mathbf{A}\mathbf{X}^{(k)}\mathbf{W}$.

**MPNNs**  Finally, we note that the Beltrami flow amounts to linear aggregation with non-linear coefficients, or the 'attentional' flavor of GNNs [10]. The more general message passing flavor [30] is possible using a generic non-linear equation of the form $\frac{\partial}{\partial t}\mathbf{Z}(t) = \mathbf{\Psi}(\mathbf{Z}(t))$.

## 4  BLEND: Beltrami Neural Diffusion

Beltrami Neural Diffusion (BLEND) is a novel class of graph neural network architectures derived from the graph Beltrami framework. We assume an input graph $\mathcal{G} = (\mathcal{V}, \mathcal{E})$ with $n$ nodes and $d$-dimensional node-wise features represented as a matrix $\mathbf{X}_{\text{in}}$. We further assume a $d'$-dimensional positional encoding $\mathbf{U}_{\text{in}}$ of the graph nodes. BLEND architectures implement a learnable joint diffusion process of $\mathbf{U}$ and $\mathbf{X}$ and runs it for time $T$, to produce an output node embeddings $\mathbf{Y}$,

$$\mathbf{Z}(0) = (\phi(\mathbf{U}_{\text{in}}), \psi(\mathbf{X}_{\text{in}})) \qquad \mathbf{Z}(T) = \mathbf{Z}(0) + \int_0^T \frac{\partial \mathbf{Z}(t)}{\partial t} \mathrm{d}t \qquad \mathbf{Y} = \xi(\mathbf{Z}(T)),$$

where $\phi$, $\psi$ are learnable positional and feature encoders and $\xi$ is a learnable decoder (possibly changing the output dimensions). Here the $\alpha$ in Equations (2) and (8) is absorbed by $\psi$ and made

learnable. $\frac{\partial \mathbf{Z}(t)}{\partial t}$ is given by the graph Beltrami flow equation (8), where the diffusivity function (attention) $a$ is also learnable. The choice of attention function depends on the geometry of the positional encoding and for Euclidean encodings we find the scaled dot product attention [86] performs well, in which case

$$a(\mathbf{z}_i, \mathbf{z}_j) = \mathrm{softmax}\left(\frac{(\mathbf{W}_K\mathbf{z}_i)^\top \mathbf{W}_Q\mathbf{z}_j}{d_k}\right) \tag{13}$$

where $\mathbf{W}_K$ and $\mathbf{W}_Q$ are learned matrices, and $d_k$ is a hyperparameter.

## 5    Experimental results

In this section, we compare the proposed Beltrami framework to popular GNN architectures on standard node classification benchmarks and provide a detailed study of the choice of the positional encoding space. Additional experiments and implementation details, including runtimes and hyperparameter tuning are given in the Supplementary Materials. The code is available at `https://github.com/twitter-research/graph-neural-pde`.

**Datasets**  In our experiments, we use the following datasets: Cora [56], Citeseer [77], Pubmed [58], CoauthorCS [79], Amazon, Computer, and Photo [55], and OGB-arxiv [35]. Since many works using the first three datasets rely on the Planetoid splits [96], we included them Table 2, together with a more robust evaluation on 100 random splits with 20 random initialisations [79].

**Baselines**  We compare to the following GNN architectures: GCN [42], GAT [88], MoNet [57] and GraphSAGE [32], and recent ODE-based GNN models: CGNN [94], GDE [67], GODE [98], and two versions of LanczosNet [49]. We use two variants of our method: using fixed input graph (BLEND) and using $k$NN graph in the positional coordinates (BLEND-knn).

### 5.1    Node Classification

In these experiments, we followed the methodology of [79] using 20 random weight initialisations for datasets with fixed Planetoid splits and 100 random splits otherwise. Where available, results from [79] are reported. Hyperparameters with the highest validation accuracy were chosen and results are reported on a test set that is used only once. Hyperparameter search used Ray Tune [50] with a thousand random trials using an asynchronous hyperband scheduler with a grace period and half life of ten epochs. The code to reproduce our results is included with the submission and will be released publicly following the review process. Experiments ran on AWS p2.8xlarge machines, each with 8 Tesla V100-SXM2 GPUs.

| Method | CORA | CiteSeer | PubMed |
|---|---|---|---|
| *GCN* | 81.9±0.8 | 69.5±0.9 | 79.0±0.5 |
| *GAT* | 82.8±0.5 | 71.0±0.6 | 77.0±1.3 |
| *MoNet* | 82.2±0.7 | 70.0±0.6 | 77.7±0.6 |
| *GS-maxpool* | 77.4±1.0 | 67.0±1.0 | 76.6±0.8 |
| *Lanczos* | 79.5±1.8 | 66.2±1.9 | 78.3±0.3 |
| *AdaLanczos* | 80.4±1.1 | 68.7±1.0 | 78.1±0.4 |
| *CGNN* | 81.7±0.7 | 68.1±1.2 | 80.2±0.3 |
| *GDE* | **83.8**±0.5 | 72.5±0.5 | 79.9±0.3 |
| *GODE* | 83.3±0.3 | 72.4±0.6 | 80.1±0.3 |
| *BLEND* | **84.2**±0.6 | **74.4**±0.7 | **80.7**± 0.7 |
| *BLEND-kNN* | 83.1±0.8 | **73.3**±0.9 | **81.5**±0.5 |

Table 2: Performance (test accuracy±std) of different GNN models using Planetoid splits.

**Implementation details**  For all datasets excepting ogb-arxiv, adaptive explicit Dormand-Prince scheme was used as the numerical solver; for ogb-arxiv, we used the Runge-Kutta method. For the two smallest datasets (Cora and Citeseer) we performed direct backpropagation through each step of the numerical integrator. For the larger datasets, to reduce memory complexity, we use Pontryagin's maximum principle to propagate gradients backwards in time [69]. For the larger datasets, kinetic energy and Jacobian regularisation [28, 37] was employed. The regularisation ensures the learned dynamics is well-conditioned and easily solvable by a numeric solver, which reduced training time. We use constant initialisation for the attention weights, $\mathbf{W}_K, \mathbf{W}_Q$, so training starts from a well-conditioned system that induces small regularisation penalty terms [28].

The space complexity of BLEND is dominated by evaluating attention (13) over edges and is $\mathcal{O}(|\mathcal{E}'|(d + d'))$ where $\mathcal{E}'$ is the edge set following rewiring and $d$ is dimension of features and $d'$ is the dimension of positional encoding. The runtime complexity is $\mathcal{O}(|\mathcal{E}'|(d + d'))(E_b + E_f)$, split

| Method | CORA | CiteSeer | PubMed | Coauthor CS | Computer | Photo | ogb-arxiv |
|---|---|---|---|---|---|---|---|
| *GCN* | 81.5±1.3 | 71.9±1.9 | 77.8±2.9 | 91.1±0.5 | 82.6±2.4 | 91.2±1.2 | 71.74±0.29 |
| *GAT* | 81.8±1.3 | 71.4±1.9 | 78.7±2.3 | 90.5±0.6 | 78.0±19 | 85.7±20 | **73.01**±0.19* |
| *GAT-ppr* | 81.6±0.3 | 68.5±0.2 | 76.7±0.3 | 91.3±0.1 | 85.4±0.3 | 90.9±0.3 | — |
| *MoNet* | 81.3±1.3 | 71.2±2.0 | 78.6±2.3 | 90.8±0.6 | 83.5±2.2 | 91.2±2.3 | — |
| *GS-mean* | 79.2±7.7 | 71.6±1.9 | 77.4±2.2 | 91.3±2.8 | 82.4±1.8 | 91.4±1.3 | 71.49±0.27 |
| *GS-maxpool* | 76.6±1.9 | 67.5±2.3 | 76.1±2.3 | 85.0±1.1 | — | 90.4±1.3 | — |
| *CGNN* | 81.4±1.6 | 66.9±1.8 | 66.6±4.4 | **92.3**±0.2 | 80.3±2.0 | 91.4±1.5 | 58.70±2.50 |
| *GDE* | 78.7±2.2 | 71.8±1.1 | 73.9±3.7 | 91.6±0.1 | 82.9±0.6 | 92.4±2.0 | 56.66±10.9 |
| ***BLEND*** | **84.8**±0.9 | **75.9**±1.3 | **79.5**±1.4 | **92.9**±0.2 | **86.9**±0.6 | **92.9**±0.6 | **72.56**±0.1 |
| ***BLEND-kNN*** | **82.5**±0.9 | **73.4**±0.5 | **80.9**±0.7 | **92.3**±0.1 | **86.7**±0.6 | **93.5**±0.3 | —† |

Table 3: Performance (test accuracy±std) of different GNN models using random splits. *OGB GAT reference has 1.5M parameters vs ours 70K.† BLEND-kNN pre-processes the graph using the DIGL methodology [43] (Section 3.3), which constructs an n-dimensional representation of each node (an n-by-n matrix), then sparsifies into a kNN graph. The ogb-arxiv dataset has >150K nodes and goes OOM. This is not a limitation of BLEND, but that of DIGL. Other forms of initial rewiring are possible, but we chose to compare with DIGL (arguably the most popular graph rewiring) and so this result is missing

between the forward and backward pass and can be dominated by either depending on the number of function evaluations ($E_b$, $E_f$).

Tables 2–3 summarise the results of our experiments. BLEND outperforms other GNNs in most of the experiments, showing state-of-the-art results on some datasets. Another important point to note is that compared GNNs use different sets of parameters per layer, whereas in BLEND, due to our choice of a time-independent attention, parameters are shared. This results in significantly fewer parameters: for comparison, the OGB versions of GCN, SAGE and GAT used in the ogb-arxiv experiment require 143K, 219K and 1.63M parameters respectively, compared to only 70K in BLEND.

## 5.2 Positional encoding

In the second experiment, we investigated the impact of the positional encodings and vary the dimensionality and underlying geometry, in order to showcase the flexibility of our framework.

Figure 3 shows that for all datasets BLEND is superior to a Euclidean model where positional encodings are not used, which corresponds to $Z = X$ (BLEND w/o positional in Figure 3) and a version of GAT where attention is over a concatenation of the same positional encodings used in BLEND and the features. The only exception is CoathorCS, where the performance without positional encodings is unchanged.

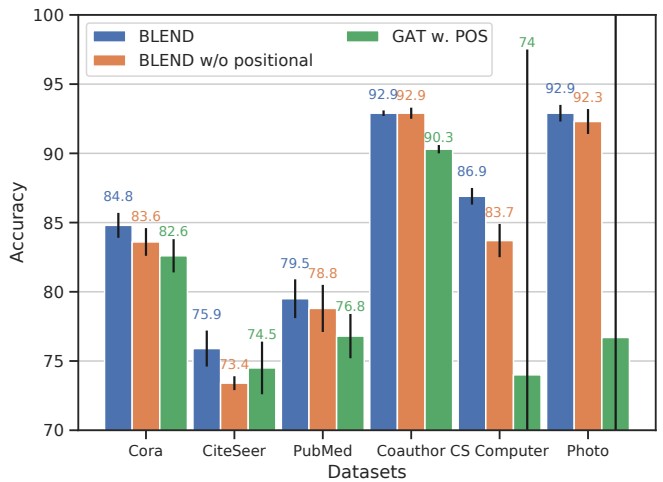

Figure 3: An ablation study showing BLEND with and without positional encodings as well as GAT, the most similar conventional GNN with positional encodings added

We experimented with three forms of positional encoding: *DIGL* PPR embeddings of dimension $n$ [43], *DeepWalk* embeddings [66] of dimensions 16–256, and *hyperbolic* embeddings in the Poincare ball [15, 59] of dimension 2–16. Positional encodings are calculated as a preprocessing step and input as the **U** coordinates to BLEND. We calculated DeepWalk node embeddings using PyTorch Geometric's Node2Vec implementation with parameters $p = 1$, $q = 1$, context size 20, and 16 walks per node. We trained with walk length ranging between 40 and 120 and took the embeddings with the highest accuracy for the link prediction task. Shallow hyperbolic embeddings were generated using the HGCN [17] implementation with the default parameters provided by the authors.

Figure 4 (left) compares the performance of the best model using DIGL $n$-dimensional positional encodings against the best performing $d$-dimensional hyperbolic positional encodings with $d$ tuned over the range 2-16. The average performance with DIGL positional encodings is 85.48, compared to 85.28 for hyperbolic encodings. Only in one case the DIGL encodings outperform the best hyperbolic

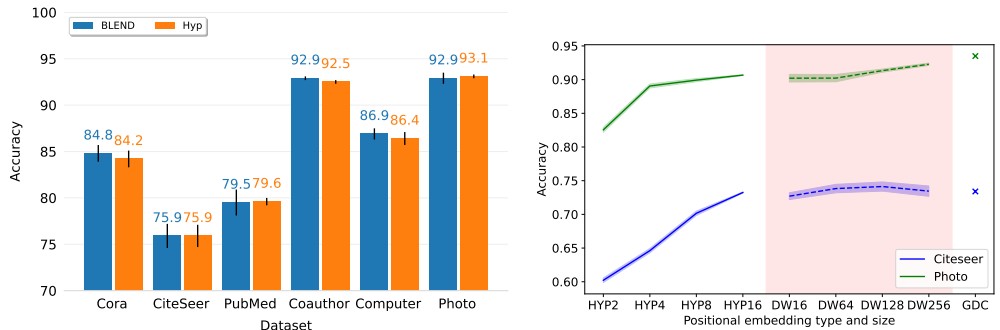

Figure 4: Left: performance comparison between Euclidean and hyperbolic positional embeddings. Right: results of positional embeddings ablation. Hyperbolic or Euclidean embeddings with $d' = 16$ allow to obtain performances comparable to euclidean embeddings with $d' = n >> 16$.

encodings. Figure 4 (right) show the change in performance with the dimension $d'$ of the positional encoding using a fixed hyperparameter configuration. As expected, we observe monotonic increase in performance as function of $d'$. Importantly most of the performance is captured by either 16 dimensional hyperbolic or positional encodings, with a small additional margin gained for going up to $d' = n$, which is impractical for larger graphs.

## 5.3 Additional ablations

In addition to studying the affect of positional encodings , we performed ablation studies on the step size used in the integrator as well as different forms of attention.

In Figure 5 we studied the affect of changing the step size of the integrator using the explicit Euler method with a fixed terminal time set to be the optimal terminal time. The left hand side of the figure shows the performance using the adaptive stepsize Dopri5 for comparison. Dopri5 gives the most consistent performance and is best if three of the six datasets. Details of additional attention functions and their relative performance can be found in the Supplementary Material.

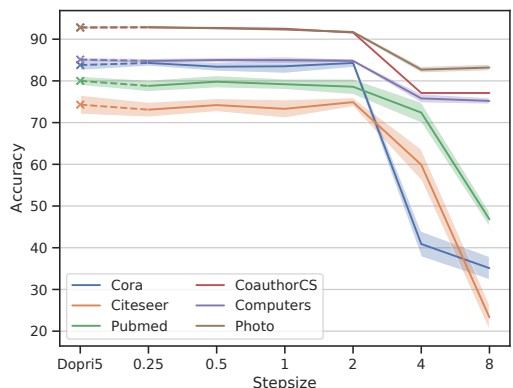

Figure 5: step size against accurcy for explicit Euler compared to the adaptive dopri5.

## 6   Related work

**Image processing, computer vision, and graphics.**   Following the Perona-Malik scheme [65], PDE-based approaches flourished in the 1990s with multiple versions of anisotropic [90] and non-Euclidean [82] diffusion used primarily for image denoising. The realisation of these ideas in the form of nonlinear filters [85, 12] was adopted in the industry. PDE-based methods were also used for low-level vision tasks including inpainting [6] and image segmentation [13, 18]. In computer graphics, fundamental solutions ('heat kernels') of diffusion equations were used as shape descriptors [83]. The closed-form expression of such solutions using the Laplace-Beltrami operator served as inspiration for some of the early approaches for GNNs [34, 23, 42, 45].

**Neural differential equations**   The interpretation of neural networks as discretised differential equations ('neural ODEs') [19] was an influential recent result with multiple follow-up works [25, 28, 53, 46]. In deep learning on graphs, this mindset has been applied to GNN architectures [2, 67] and continuous message passing [94]. Continuous GNNs were also explored in [31] who, similarly to [76], addressed the solutions of fixed point equations. Ordinary Differential Equations

on Graph Networks (GODE)[98] approach the problem using the technique of invertible ResNets. Finally, [75] used graph-based ODEs to generate physics simulations.

**Physics-inspired learning**    Solving PDEs with deep learning has been explored by [72]. Neural networks appeared in [47] to accelerate PDE solvers with applications in the physical sciences. These have been applied to problems where the PDE can be described on a graph [48]. [4] consider the problem of predicting fluid flow and use a PDE inside a GNN. These approaches differ from ours in that they solve a given PDE, whereas we use the notion of discretising PDEs as a principle to understand and design GNNs.

**Neural ODEs on graphs**    The most similar work to this is GRAND [16] of which BLEND can be considered a non-Euclidean extension. In addition there are several other works that apply the neural ODE framework to graphs. In GDE [67], GODE [98] and CGNN [94], the goal is to adapt neural ordinary differential equations to graphs. In contrast, we consider non-Euclidean partial differential equations and regard GNNs as particular instances of their discretisation (both spatial and temporal). We can naturally use spaces with any metric, in particular, extending recent works on hyperbolic graph embeddings. None of the previous techniques explore the link to differential geometry. More specifically, GDE, GODE, and CGNN consider Neural ODEs of the canonical form $\frac{\partial x}{\partial t} = f(x, t, \theta)$ where $f$ is a graph neural network (GODE), the message passing component of a GNN (CGNN), or restricting $f$ to be layers of bijective functions on graphs (GDE). Furthermore, in CGNN only ODEs with closed form solutions are considered. [75] on the other hand is quite distinct as they are not concerned with GNN design and instead use graph-based ODEs to generate physics simulations.

## 7    Conclusion

We developed a new class of graph neural networks based on the discretisation of a non-Euclidean diffusion PDE called Beltrami flow. We represent the graph structure and node features as a manifold in a joint continuous space, whose evolution by a parametric diffusion PDE (driven by the downstream learning task) gives rise to feature learning, positional encoding, and possibly also graph rewiring. Our experimental results show very good performance on popular benchmarks with a small fraction of parameters used by other GNN models. Perhaps most importantly, our framework establishes important links between GNNs, differential geometry, and PDEs – fields with a rich history and profound results that in our opinion are still insufficiently explored in the graph ML community.

**Future directions**    While we show that our framework generalises many popular GNN architectures we seek to use the graph as a numerical representation of an underlying continuous space. This view of the graph as an approximation of a continuous latent structure is a common paradigm of manifold learning [84, 5, 20] and network geometry [8]. If adopted in graph ML, this mindset offers a rigorous mathematical framework for formalising and generalising some of the recent trends in the field, including the departure from the input graph as the basis for message passing [1], latent graph inference [41, 89, 29, 36] higher-order [7, 54] and directional [3] message passing (which can be expressed as anisotropic diffusion arising from additional structure of the underlying continuous space and different discretisation of the PDEs e.g. based on finite elements), and exploiting efficient numerical PDE solvers [19]. We leave these exciting directions for future research.

**Societal impact**    GNNs have recently become increasingly utilized in industrial applications e.g. in recommender systems and social networks, and hence could potentially lead to a negative societal impact if used improperly. We would like to emphasize that our paper does not study such potential negative applications and the mathematical framework we develop could help to interpret and understand existing GNN models. We believe that better understanding of ML models is key to managing their potential societal implications and preventing their negative impact.

**Limitations**    The assumption that the graph can be modelled as a discretisation of some continuous space makes our framework applicable only to cases where the edge and node features are continuous in nature. Applications e.g. to knowledge graphs with categorical attributes could only be addressed by first embedding such attributes in a continuous space. Finally, the structural result presented in Theorem 1 that link our model to the Polyakov action have not been implemented. This will be addressed in future works.

## 8 Acknowledgements

We thank Nils Hammerla and Gabriele Corso for feedback on early version of this manuscript. MB and JR are supported in part by ERC Consolidator grant no 724228 (LEMAN).

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
