# Supplementary Material: Beltrami Flow and Neural Diffusion on Graphs

**Benjamin P. Chamberlain**[*]
Twitter Inc.
bchamberlain@twitter.com

**James Rowbottom**[*]
Twitter Inc.

**Davide Eynard**
Twitter Inc.

**Francesco Di Giovanni**
Twitter Inc.

**Xiaowen Dong**
University of Oxford

**Michael M. Bronstein**
Twitter Inc. and Imperial College London

## A   Additional experimental results and implementation details

**Datasets**   The statistics for the largest connected components of the experimental datasets are given in Table 1.

| Dataset | Type | Classes | Features | Nodes | Edges | Label rate |
|---------|------|---------|----------|-------|-------|------------|
| Cora | citation | 7 | 1433 | 2485 | 5069 | 0.056 |
| Citeseer | citation | 6 | 3703 | 2120 | 3679 | 0.057 |
| PubMed | citation | 3 | 500 | 19717 | 44324 | 0.003 |
| Coauthor CS | co-author | 15 | 6805 | 18333 | 81894 | 0.016 |
| Computers | co-purchase | 10 | 767 | 13381 | 245778 | 0.015 |
| Photos | co-purchase | 8 | 745 | 7487 | 119043 | 0.021 |
| OGB-Arxiv | citation | 40 | 128 | 169343 | 1166243 | 0.005 |

Table 1: Dataset Statistics

**Replication of results and hyper-parameters**   Code to regenerate our experimental results together with hyperparameters for all datasets is provided. The hyperparameters are listed in `best_params.py`. Refer to the `README.md` for instructions to run the experiments.

**Numerical ODE solver**   We use the library `torchdiffeq` [2] to discretise the continuous time evolution and learn the system dynamics. The Pontryagin maximum / adjoint method is used to replace backpropagation for all datasets, with the exception of Cora and Citeseer due to the high memory complexity of applying backpropagation directly through the computational graph of the numerical integrator.

**Decoupling the terminal integration time between inference and training**   At training time we use a fixed terminal time $T$ that is tuned as a hyperparameter. For inference, we treat $T$ as a flexible parameter to which we apply a patience and measure the validation performance throughout the integration.

**Data splits**   We report transductive node classification results in Tables 2 and 3 in the main paper. To facilitate direct comparison with many previous works, Table 2 uses the Planetoid splits given by [8]. As discussed in e.g. [7], there are many limitations with results based on these fixed dataset splits and so we report a more robust set of experimental results in Table 3. Here we use 100 random splits with 20 random weight initializations. In each case 20 labelled nodes per class are used at training time, with the remaining labels split between test and validation. This methodology is consistent with [7].

35th Conference on Neural Information Processing Systems (NeurIPS 2021), Sydney, Australia.

| Method | Cora | Citeseer | Pubmed | CoauthorCS | Computers | Photo |
|--------|------|----------|--------|------------|-----------|-------|
| cosine_sim | $83.8 \pm 0.5$ | $73.5 \pm 1.1$ | $79.0 \pm 2.3$ | $92.8 \pm 0.1$ | $84.8 \pm 0.7$ | $93.2 \pm 0.4$ |
| exp_kernel | $83.6 \pm 1.8$ | $75.0 \pm 1.3$ | $79.9 \pm 1.4$ | $92.8 \pm 0.2$ | $84.8 \pm 0.5$ | $93.2 \pm 0.5$ |
| pearson | $83.7 \pm 1.5$ | $75.2 \pm 1.3$ | $\mathbf{80.0 \pm 1.2}$ | $92.8 \pm 0.2$ | $84.8 \pm 0.5$ | $\mathbf{93.5 \pm 0.3}$ |
| scaled dot | $\mathbf{84.8 \pm 0.9}$ | $\mathbf{75.9 \pm 1.3}$ | $79.5 \pm 1.4$ | $\mathbf{92.9 \pm 0.2}$ | $\mathbf{86.9 \pm 0.6}$ | $92.9 \pm 0.6$ |

Table 2: Performance with different attention functions. Scaled dot is best performing in four of the six experiments

**Positional encodings** In section 5.2 of the main paper three types of positional encoding are described: *DIGL PPR* [4], *DeepWalk* [6] and *hyperbolic* embeddings [1, 5]. We provide the latter two as preprocessed pickle files within our repo. The DIGL PPR positional encodings are too large and so these are automatically generated and saved whenever code that requires them is run.

**Diffusivity (attention) function** In section 4 of the main paper one choice of attention function is described. Additionally four alternatives were considered, which achieved roughly equivalent performance:

- Scaled dot

$$a(\mathbf{z}_i, \mathbf{z}_j) = \text{softmax}\left(\frac{(\mathbf{W}_K \mathbf{z}_i)^\top \mathbf{W}_Q \mathbf{z}_j}{d_k}\right) \qquad (1)$$

- Cosine similarity

$$a(\mathbf{z}_i, \mathbf{z}_j) = \text{softmax}\left(\frac{(\mathbf{W}_K \mathbf{z}_i)^\top \mathbf{W}_Q \mathbf{z}_j}{\|\mathbf{W}_K \mathbf{z}_i\|\|\mathbf{W}_Q \mathbf{z}_j\|}\right) \qquad (2)$$

- Pearson correlation

$$a(\mathbf{z}_i, \mathbf{z}_j) = \text{softmax}\left(\frac{(\mathbf{W}_K \mathbf{z}_i - \overline{\mathbf{W}_K \mathbf{z}_i})^\top (\mathbf{W}_Q \mathbf{z}_j - \overline{\mathbf{W}_Q \mathbf{z}_j})}{\|\mathbf{W}_K \mathbf{z}_i - \overline{\mathbf{W}_K \mathbf{z}_i}\|\|\mathbf{W}_Q \mathbf{z}_j - \overline{\mathbf{W}_Q \mathbf{z}_j}\|}\right) \qquad (3)$$

- Exponential kernel

$$a(\mathbf{z}_i, \mathbf{z}_j) = \text{softmax}\left((\sigma_u \sigma_x)^2 e^{-\|\mathbf{W}_K \mathbf{u}_i - \mathbf{W}_Q \mathbf{u}_j\|^2 / 2\ell_u^2} e^{-\|\mathbf{W}_K \mathbf{x}_i - \mathbf{W}_Q \mathbf{x}_j\|^2 / 2\ell_x^2}\right) \qquad (4)$$

We additionally conducted an ablation study using these functions, results are given in Table 2.

## B  Theory of Beltrami flow

This section proves Theorem 1 in the main paper:

**Theorem 1.** *Under structural assumptions on the diffusivity, graph Beltrami flow*

$$\frac{\partial \mathbf{z}_i(t)}{\partial t} = \sum_{j:(i,j)\in\mathcal{E}'} a(\mathbf{z}_i(t), \mathbf{z}_j(t))(\mathbf{z}_j(t) - \mathbf{z}_i(t)) \qquad \mathbf{z}_i(0) = \mathbf{z}_i; \quad i = 1, \dots, n; \quad t \geq 0 \qquad (5)$$

*is the gradient flow of the discrete Polyakov functional.*

### B.1  Polyakov action and Beltrami flow on manifolds

In this section we briefly review the analysis in [3] to motivate the introduction of a discrete Polyakov action on graphs. Assume that $(\Sigma, g)$ and $(M, h)$ are Riemannian manifolds with coordinates $\{x_\mu\}$ and $\{y_\ell\}$ respectively. The *Polyakov action* for an embedding $Z : (\Sigma, g) \to (M, h)$ can be written as

$$S[Z, g, h] = \sum_{\mu,\nu=1}^{\dim(\Sigma)} \sum_{\ell,m=1}^{\dim(M)} \int_\Sigma h_{\ell m}(Z) \partial_\mu Z^\ell \partial_\nu Z^m g^{\mu\nu} \, d\text{vol}(g),$$

with $d\mathrm{vol}(g)$ the volume form on $\Sigma$ associated with the metric $g$. We restrict to the case where $(M, h)$ is the $d$-dimensional Euclidean space $(\mathbb{R}^d, \delta)$, so that the functional becomes

$$S[Z, g] = \sum_{\mu,\nu=1}^{\dim(\Sigma)} \sum_{\ell=1}^{d} \int_{\Sigma} \partial_\mu Z^\ell \partial_\nu Z^\ell g^{\mu\nu} \, d\mathrm{vol}(g).$$

The quantity above can be rewritten more geometrically as

$$S[Z, g] = \sum_{\ell=1}^{d} \int_{\Sigma} ||\nabla_g Z^\ell||_g^2 \, d\mathrm{vol}(g). \tag{6}$$

From equation (6) we see that the Polyakov action is measuring the smoothness of the embedding - more precisely its Dirichlet energy with respect to the metric $g$ - on each feature channel $\ell = 1, \ldots, d$. Given a geometric object $\Sigma$, it is natural to find an optimal way of mapping it to a larger space, in this case $(\mathbb{R}^d, \delta)$. Accordingly, one can minimize the functional in (6) either with respect to the embedding $Z$ or with respect to both the embedding $Z$ and the metric $g$. If we choose to minimize $S$ by varying both $Z$ and $g$ we find that the metric $g$ must be the metric induced on $\Sigma$ by the map $Z$, namely its pullback. In local coordinates, this amounts to the constraint below:

$$g_{\mu\nu} = \sum_{\ell=1}^{d} \partial_\mu Z^\ell \partial_\nu Z^\ell. \tag{7}$$

Once the condition on $g$ is satisfied, the Euler-Lagrange equations for each feature channel become

$$\sum_{\mu,\nu} \frac{1}{\sqrt{\det(g)}} \partial_\mu(\sqrt{\det(g)} g^{\mu\nu} \partial_\nu Z^\ell) = \mathrm{div}_g \nabla_g Z^\ell = \Delta_g Z^\ell = 0, \tag{8}$$

where the operator $\Delta_g$ is the *Laplace-Beltrami* operator on $\Sigma$ associated with the metric $g$. As a specific example, if we embed an image $\Sigma \subset \mathbb{R}^2$ into $\mathbb{R}^3$ via a grey color mapping $I \times x : (u_1, u_2) \mapsto (u_1, u_2, x(u_1, u_2))$, the gradient flow associated with the functional $S$ and hence the Euler-Lagrange equation (8) for the grey channel is exactly the one reported in Section 2.

Therefore, from a high-level perspective, the Beltrami flow derived from the Polyakov action consists in minimizing a functional $S$ representing the Dirichlet energy of the embedding $Z$ computed with respect to a metric $g$ depending - precisely via the pullback - on the embedding itself.

### B.2 A discrete Polyakov action

We consider the graph counterpart to the problem of minimizing a generalized Dirichlet energy with respect to the embedding. Let $\mathcal{G} = (\mathcal{V}, \mathcal{E})$ be a simple, undirected and unweighted graph with $|V| = n$. Also let $L^2(\mathcal{V})$ and $L^2(\mathcal{E})$ denote the space of signal functions $y : \mathcal{V} \to \mathbb{R}$ and $Y : \mathcal{E} \to \mathbb{R}$ respectively. Recall that the graph gradient of $y \in L^2(\mathcal{V})$ is defined by

$$\nabla_\mathcal{G} y \in L^2(\mathcal{E}) : (i, j) \mapsto y_j - y_i.$$

Its adjoint operator is called graph divergence and satisfies

$$\mathrm{div}_\mathcal{G} Y \in L^2(\mathcal{V}) : i \mapsto \sum_{j:(i,j)\in\mathcal{E}} Y_{ij}.$$

Assume now we have a graph embedding $\mathbf{Z} : \mathcal{V} \to \mathbb{R}^{d'} \times \mathbb{R}^d$ of the form $\mathbf{z}_i = (\mathbf{u}_i, \alpha \mathbf{x}_i)$ for some scaling $\alpha \geq 0$, with $1 \leq i \leq n$. The coordinates $\mathbf{u}_i$ and $\mathbf{x}_i$ are called *positional* and *feature* coordinates respectively. In analogy with the Polyakov action (6), we introduce a modified Dirichlet energy measuring the smoothness of a graph embedding $\mathbf{Z}$ across its different channels: given a family of *nonnegative* maps $(\psi_{ij}^\ell)$, with $\psi_{ij}^\ell : \mathbb{R}^{n\times(d'+d)} \to \mathbb{R}$, $1 \leq i, j \leq n$ and $1 \leq \ell \leq d' + d$, we define

$$S[\mathbf{Z}, \psi] := \frac{1}{2} \sum_{\ell=1}^{d'+d} \sum_{i,j=1}^{n} A_{ij} \psi_{ij}^\ell(\mathbf{Z}), \tag{9}$$

where $A$ is the graph adjacency matrix. Defining the following $\psi$-weighted norm for each $\ell \in [1, d' + d]$

$$||\nabla_i z^\ell||^2_\psi := \sum_{j=1}^n A_{ij} \psi^\ell_{ij}(\mathbf{Z}),$$

then (9) can be written as

$$S[\mathbf{Z}, \psi] = \frac{1}{2} \sum_{\ell=1}^{d'+d} \sum_{i=1}^n ||\nabla_i z^\ell||^2_\psi. \tag{10}$$

Beyond the notational similarity with (6), the quantity $S[\mathbf{Z}, \psi]$ sums the integrals over all channels $1 \le \ell \le d' + d$ - i.e. summations on the vertex set - of the norms of the gradients of $z^\ell$ exactly as for the continuum Polyakov action. The dependence of such norms on the embedding is to take into account that in the smooth case the Beltrami flow imposes the constraint (7), meaning that the metric $g$ with respect to which we compute the gradient norm of the embedding depends on the embedding itself. We note that the choice

$$\psi^\ell_{ij}(\mathbf{Z}) = (z^\ell_j - z^\ell_i)^2 = (\nabla_\mathcal{G} z^\ell(i,j))^2, \tag{11}$$

yields the classic Dirichlet energy of a multi-channel graph signal

$$S[\mathbf{Z}] = \frac{1}{2} \sum_{\ell=1}^{d'+d} \sum_{i,j=1}^n A_{ij}(\nabla_\mathcal{G} z^\ell(i,j))^2,$$

From now on we refer to the function $S : (\mathbf{Z}, \psi) \mapsto S[\mathbf{Z}, \psi]$ in (9) as the *discrete Polyakov action*.

### B.3 Proof of Theorem 1

We recall that given an embedding $\mathbf{Z} : \mathcal{V} \to \mathbb{R}^{d'+d}$, we are interested in studying a discrete diffusion equation of the form

$$\frac{\partial \mathbf{z}_i(t)}{\partial t} = \sum_{j:(i,j)\in\mathcal{E}} a(\mathbf{z}_i(t), \mathbf{z}_j(t))(\mathbf{z}_j(t) - \mathbf{z}_i(t)) \qquad \mathbf{z}_i(0) = \mathbf{z}_i; \quad i = 1, \dots, n; \quad t \ge 0. \tag{12}$$

The coupling $a(\mathbf{z}_i(t), \mathbf{z}_j(t))$ is called *diffusivity*. We now prove that the differential system above is the gradient flow of the discrete Polyakov action $S[\mathbf{Z}, \psi]$ under additional assumptions on the structure of the diffusivity. We restate Theorem 1 in a more precise way:

**Theorem 1.** *Let $(\psi^\ell_{ij})$ be a family of maps $\psi^\ell_{ij} : \mathbb{R}^{n \times (d'+d)} \to \mathbb{R}$ satisfying the assumptions*

(i) *There exist a family of maps $(\tilde{\psi}_{ij})$, with $\tilde{\psi}_{ij} : \mathbb{R}^n \to \mathbb{R}$, such that*

$$\psi^\ell_{ij}(\mathbf{Z}) = \tilde{\psi}_{ij} \left( ||\mathbf{z}_i - \mathbf{z}_1||^2, \dots, ||\mathbf{z}_i - \mathbf{z}_n||^2 \right) (z^\ell_j - z^\ell_i)^2. \tag{13}$$

(ii) *If we write $\tilde{\psi}_{ij} : (p_1, \dots, p_n) \mapsto \tilde{\psi}_{ij}(p_1, \dots, p_n)$, then we require*

$$\partial_{p_k} \tilde{\psi}_{ij}(\mathbf{p}) = 0, \quad if \ (i,k) \notin \mathcal{E}.$$

*Then, the gradient flow associated with $S[\mathbf{Z}, \psi]$ is given by (12), with the diffusivity $a$ satisfying*

$$a(\mathbf{z}_i(t), \mathbf{z}_j(t)) = (\tilde{\psi}_{ij} + \tilde{\psi}_{ji})(\mathbf{Z}(t))$$
$$+ \sum_{k:(i,k)\in\mathcal{E}} \partial_j \tilde{\psi}_{ik}(\mathbf{Z}(t)) ||\mathbf{z}_k(t) - \mathbf{z}_i(t)||^2 + \sum_{k:(j,k)\in\mathcal{E}} \partial_i \tilde{\psi}_{jk}(\mathbf{Z}(t)) ||\mathbf{z}_k(t) - \mathbf{z}_j(t)||^2$$

*where the dependence of $\tilde{\psi}$ on $\mathbf{Z}(t)$ is as in (13).*

**Remark.** *We observe that by taking $\tilde{\psi}_{ij}$ to be the constant function one, we recover the classical case (11). In fact, for such choice the diffusivity satisfies $a(\mathbf{z}_i(t), \mathbf{z}_j(t)) = 2$ and the gradient flow is given by the graph Laplacian, namely*

$$\frac{\partial z^\ell_i(t)}{\partial t} = (2\Delta z^\ell(t))_i.$$

*Proof.* Once we choose the family of maps $\psi_{ij}^\ell$ as in the statement, the discrete Polyakov action is a map $S : \mathbb{R}^{n \times (d'+d)} \to \mathbb{R}$. To ease the notation, given a vector $\mathbf{Z} \in \mathbb{R}^{n \times (d'+d)}$ we write $\mathbf{Z} = (z_1^1, \ldots, z_n^1, \ldots, z_1^{d'+d}, \ldots, z_n^{d'+d})$. To prove the result we now simply need to compute the gradient of the functional $S[\mathbf{Z}]$: given $1 \le r \le d' + d$ and $1 \le k \le n$ we have

$$\frac{\partial S[\mathbf{Z}]}{\partial z_k^r} = \frac{1}{2} \sum_{\ell=1}^{d'+d} \sum_{i,j=1}^n A_{ij} \partial_{z_k^r} \psi_{ij}^\ell(\mathbf{Z}). \tag{14}$$

From the assumptions we can expand the partial derivative of the map $\psi_{ij}^\ell$ as

$$\partial_{z_k^r} \psi_{ij}^\ell(\mathbf{Z}) = 2 \sum_{s=1}^n \sum_{q=1}^{d'+d} \partial_s \tilde{\psi}_{ij}(\mathbf{Z})(z_i^q - z_s^q)\delta_{rq}(\delta_{ik} - \delta_{sk})(z_j^\ell - z_i^\ell)^2 + 2\tilde{\psi}_{ij}(\mathbf{Z})(z_j^\ell - z_i^\ell)\delta_{\ell r}(\delta_{jk} - \delta_{ik}),$$

where we have simply written $\tilde{\psi}_{ij}\left(||\mathbf{z}_i - \mathbf{z}_1||^2, \ldots, ||\mathbf{z}_i - \mathbf{z}_n||^2\right) = \tilde{\psi}_{ij}(\mathbf{Z})$. Therefore, we can write (14) as

$$\begin{aligned}
\frac{\partial S[\mathbf{Z}]}{\partial z_k^r} =\ & \sum_{j,s=1}^n A_{kj}(\partial_s \tilde{\psi}_{kj}(\mathbf{Z}))(z_k^r - z_s^r)||\mathbf{z}_j - \mathbf{z}_k||^2 \\
& + \sum_{i,j=1}^n A_{ij}(\partial_k \tilde{\psi}_{ij}(\mathbf{Z}))(z_k^r - z_i^r)||\mathbf{z}_j - \mathbf{z}_i||^2 \\
& + \sum_{j=1}^n A_{kj}(\tilde{\psi}_{kj} + \tilde{\psi}_{jk})(\mathbf{Z}))(z_k^r - z_j^r).
\end{aligned}$$

We now use the assumption (ii) to see that $\partial_s \tilde{\psi}_{kj} = A_{ks}\partial_s \tilde{\psi}_{kj}$ and similarly for $\partial_k \tilde{\psi}_{ij} = A_{ki}\partial_k \tilde{\psi}_{ij}$. Up to renaming dummy indices we get

$$\frac{\partial S[\mathbf{Z}]}{\partial z_k^r} = \sum_{j:(j,k)\in\mathcal{E}} \left((\tilde{\psi}_{kj} + \tilde{\psi}_{jk})(\mathbf{Z}) + \sum_{p:(k,p)\in\mathcal{E}}\partial_j \tilde{\psi}_{kp}(\mathbf{Z})||\mathbf{z}_p - \mathbf{z}_k||^2 + \sum_{p:(j,p)\in\mathcal{E}}\partial_k \tilde{\psi}_{jp}(\mathbf{Z})||\mathbf{z}_p - \mathbf{z}_j||^2\right)(z_k^r - z_j^r).$$

By inspection, we see that the right hand side can be rewritten as

$$\frac{\partial S[\mathbf{Z}]}{\partial z_k^r} = \sum_{j:(j,k)\in\mathcal{E}} a(\mathbf{z}_k, \mathbf{z}_j)(z_k^r - z_j^r),$$

with $a$ the diffusivity given in the statement of Theorem 1. Therefore, the gradient flow associated with $S[\mathbf{Z}]$ is

$$\frac{\partial z_k^r(t)}{\partial t} = -\frac{\partial S[\mathbf{Z}]}{\partial z_k^r} = \sum_{j:(j,k)\in\mathcal{E}} a(\mathbf{z}_k(t), \mathbf{z}_j(t))(z_j^r(t) - z_k^r(t)),$$

for each $1 \le k \le n$ and $1 \le r \le d' + d$. This completes the proof. $\qquad\square$

## C   Implementation Details

**Runtimes**   We include below the training and inference runtimes of BLEND and BLEND-kNN, compared with corresponding runtimes of GAT. We use a standard GAT implementation (also used to generate results in the main paper) with two layers and eight heads (all other hyperparameters are tuned). Experiments used a Tesla K80 GPU with 11GB of RAM. The relative runtimes of BLEND and BLEND-kNN are largely driven by the respective edge densities and so we also report these in the table. For the small graph BLEND-kNN significantly densifies the graphs (as is consistent with DIGL) and runtimes are longer. For the large graphs the effect is less pronounced and even reversed in the case of Computers.

Computational cost can be computed using from the runtimes based on the AWS rental cost of a Tesla K80 GPU, which is currently less than $1 / hour.

|  | Cora | Citeseer | Pubmed | CoauthorCS | Computers | Photo |
|---|---|---|---|---|---|---|
| BLEND (s) | 26.79 | 27.74 | 168.71 | 183.29 | 230.20 | 69.85 |
| BLEND_kNN (s) | 44.65 | 83.84 | 387.86 | 216.94 | 157.77 | 73.86 |
| Av. degree | 4.1 | 3.5 | 4.5 | 8.9 | 36.70 | 31.8 |
| Av. degree kNN | 32 | 64 | 61.1 | 12.4 | 22 | 38.2 |
| GAT (s) | 1.68 | 1.76 | 9.85 | 18.96 | 27.12 | 10.65 |
| BLEND / GAT | 15.95 | 15.72 | 17.12 | 9.67 | 8.49 | 6.56 |
| BLEND_kNN / GAT | 26.59 | 47.51 | 39.36 | 11.44 | 5.82 | 6.93 |

Table 3: Training times (s) for 100 epochs

|  | Cora | Citeseer | Pubmed | CoauthorCS | Computers | Photo |
|---|---|---|---|---|---|---|
| BLEND (s) | 0.0712 | 0.0878 | 0.2955 | 0.449 | 0.4603 | 0.2071 |
| BLEND_kNN (s) | 0.114 | 0.2458 | 0.9463 | 0.4945 | 0.3029 | 0.2246 |
| Av. degree | 4.1 | 3.5 | 4.5 | 8.9 | 36.7 | 31.8 |
| Av. degree kNN | 32 | 64 | 61.1 | 12.4 | 22 | 38.2 |
| GAT (s) | 0.0031 | 0.0039 | 0.0070 | 0.0326 | 0.0165 | 0.0093 |
| BLEND / GAT | 22.72 | 22.46 | 42.5 | 13.8 | 27.91 | 22.35 |
| BLEND_kNN / GAT | 36.40 | 62.88 | 136.08 | 15.19 | 18.37 | 24.23 |

Table 4: Inference times (s)

**Hyperparameter Tuning**   We tuned the neural ODE based methods using Ray Tune. The remaining results were taken from the Pitfalls of GNNs paper. As this paper applied a thorough hyperparameter search and we replicated their experimental methodology we did not feel that it was necessary to independently tune these methods.

We did not apply Jacobian or kinetic regularisation for CGNN, GODE or GDE. We used regularisation to reduce the number of function evaluations made by the solver, which made training faster and more stable, but did not improve performance.

**Softmax versus Squareplus**   In some cases we found it beneficial to replace the $\mathrm{softmax}$ operator with $\mathrm{squareplus}$ which replaces $e^x$ with $\frac{1}{2}(x + \sqrt{x^2 + 4})$. This normalises logits (like the softmax), but has a gradient approaching one for large $x$, preventing one edge from dominating the diffusivity function.