# OpenReview forum: "Beltrami Flow and Neural Diffusion on Graphs"
_NeurIPS.cc/2021/Conference — NeurIPS 2021 Poster_

### Official Review · Reviewer_RFfM · 2021-07-09

**Rating:** 7
**Confidence:** 4

**Summary:**

** score updated 6->7 after rebuttal**

The paper proposes an interpretation of various GNN related architectures as Beltrami flows on a 2-manifold of position and feature embeddings, using this perspective to formulate a GNN architecture which uses feature encoders to create starting points in the 2-manifold and learned attention kernels (in their experiments, shared across time/layers) to give the update of the embeddings. The authors propose that this unifies various architectures as well as graph-rewiring techniques developed in the literature, evalaute against GCN,GAT,GAT-ppr,MoNet,GS-{mean/maxpool},CGNN and G(O)DE on cora,citeseer,pubmed,coauthor-CS,computer,photo and ogb-arxiv, showing their method matches or outperforms all other methods when taking into account parameter efficiency.

**Limitations And Societal Impact:**

I think the authors are understating the negative effect  social network analysis can have in terms of surveillance and manipulation, as well as understating the benefit that can come from better architectures for relational tasks expressible as graphs.  VLSI,logic programming etc. can all benefit from this, as can the design of automated propaganda techniques, LAWs and police states.

As for limitations, as I said, computational expense should be discussed in the paper as well, or noted as a bottleneck.

**Main Review:**

The authors follow the NODE approach of viewing residual layers as ODE steps, allowing a continuous depth when training with differential equation solvers and parametrizing an implicit layer.

The paper is well written (except for some typos, see below) and the perspective of graph rewiring through a continuous  process is intriguing. However, I have some questions before I can give  a higher rating

-  you present various extensions in 3.4, what would be the expressive power hierarchy? I assume it goes deepsets \subset Beltrami with time independent attention(w.t.i.a)  \subset Onsager w.t.i.a \subset Onsager Onsager w.t.d.a \subset onsager with generic time dependent nonlinear equation == full attention network with enough layers?
- how does this work compare with e.g.  https://arxiv.org/abs/1909.12790 and  G(O)DE/CGNN specifically? I think it's worth highlighting commonalities and differences, especially since G(O)DE/CGNN seem competetive with your method in table2  and 3
-  since you are using attention on the node embeddings, this means this method will be quadratic in cost w.r.t to number of nodes? as opposed to the sparsity amortization afforded by e.g. MPGNN?
- you highlight the expressive power in the abstract, but if I undertand correctly, your method can only ever be equivalent to MPGNNs with rewiring techniques/full attention right? Or do you have a proof/example that says otherwise?
- did you perform tuning with raytune for *all* methods not taken from the Pitfalls of GNN paper? It might be worth marking which is taken from the paper with a cross or something, and state explicitly whether the others were tunes fairly
- following up on the complexity question, I think adding training curves or some details about training costs and parameter counts to the appendix would be beneficial to putting the work in context. in general, NODEs offer a tradeoff, not a free lunch
- did you apply the jacobian/kinetic regularisation to G(O)D/CGNN as well? how does your method perform without the regularisation?

Typos:
I assume in the equation between 212 and 213 Y is meant to be decoded with \xi, not with \psi, which is referred to as encoder in 213?


**Time Spent Reviewing:**

4

---

> ### Author Response · Authors · 2021-08-10
> **Initial Response to Reviewer RFfM**
>
> We thank the reviewer for their careful consideration and for praising the quality of our writing. We greatly appreciate them highlighting our typos and for the interesting questions, which we we have answered below.
>
> **You present various extensions in 3.4, what would be the expressive power hierarchy? I assume it goes deepsets $\subset$ Beltrami with time independent attention(w.t.i.a) $\subset$ Onsager w.t.i.a $\subset$ Onsager w.t.d.a $\subset$ Onsager with generic time dependent nonlinear equation $==$ full attention network with enough layers?**
>
> The question of the expressive power hierarchy is rather deep and requires additional research. Indeed, as the Reviewer suggests, Deepsets can be obtained as a particular case of Beltrami with a time-independent diffusivity, which is in turn a particular case of a more broader class of Onsager diffusion, and so on. However, this does not automatically imply a broader class of solutions of PDEs with such structure (hence probably the only statement that can be made is “not weaker than”).
>
> We also note that the common framework for studying the expressive power of GNNs is the Weisfeiler-Lehman hierarchy of graph isomorphism tests (k-WL hierarchy). MPNNs are equivalent to the 1-WL test. However, GNNs using graph rewiring fall outside of this hierarchy; recent works (e.g. of Bodnar et al. 2021) showed that by changing the underlying graph/message passing mechanism, one can obtain architectures strictly more powerful than WL. Since BLEND incorporates graph rewiring, we conjecture it might be made more powerful than WL, but this should be rigorously studied in future work.
>
> **Provide a detailed comparison between this and other NODE methods.**
>
> Please see our detailed reply to Reviewer eTTt **Discuss the novelty compared to other Neural ODE benchmarks**
>
> **Since you are using attention on the node embeddings, this means this method will be quadratic in cost w.r.t to the number of nodes? as opposed to the sparsity amortization afforded by e.g. MPGNN?**
> Attention is only calculated sparsely over edges of the graph, just as in GAT, and so the complexity is not quadratic in the number of nodes, but linear in the number of edges, $O(|E|)$, which is $~ O(n)$ for many real world graphs. For kNN graphs, the complexity is also $O(n)$.
>
> **You highlight the expressive power in the abstract, but if I understand correctly, your method can only ever be equivalent to MPGNNs with rewiring techniques/full attention right? Or do you have a proof/example that says otherwise?**
>
> In the literature survey in the Introduction, we indeed mention the limitations of expressive power and information bottlenecks as a motivation that has led the Graph ML community to consider variations to the traditional message passing schemes based on the modification of the underlying graph (“rewiring”) and positional encoding. We did not mean to claim that our scheme is more expressive (as we argue above, it might be, but this requires a separate study) and will reword accordingly.
>
> **Did you perform tuning with raytune for all methods not taken from the Pitfalls of GNN paper? It might be worth marking which is taken from the paper with a cross or something, and state explicitly whether the others were tunes fairly**
>
> We will add this clarification to the results table. We tuned the neural ODE based methods using Ray Tune. The remaining results were taken from the Pitfalls of GNNs paper. As this paper applied a thorough hyperparameter search and we replicated their experimental methodology we did not  feel that it was necessary to independently tune these methods.
>
> **Did you apply the jacobian/kinetic regularisation to G(O)D/CGNN as well? how does your method perform without the regularisation?**
>
> We did not apply Jacobian or kinetic regularisation for CGNN, GODE or GDE. We used regularisation to reduce the number of function evaluations made by the solver, which made training faster and more stable, but did not improve performance. An alternative to this form of regularisation is to tune the solver tolerances or to use semi-norms described in https://arxiv.org/abs/2009.09457.
>
> **Adding training curves or some details about training costs and parameter counts to the Appendix would be beneficial to putting the work in context. in general, NODEs offer a tradeoff, not a free lunch**
>
> We will add the requested details to the paper. For training times, see our response to Reviewer 3zke **Include some actual run-times of BLEND(-kNN) vs other methods**. Computational cost can be computed using from the runtimes based on the AWS rental cost of a Tesla K80 GPU, which is currently less than $1 / hour.  Parameter counts appear in Implementation Details at the end of Section 5.1. In the ogb-arxiv experiment, GCN, SAGE and GAT require 143K, 219K and 1.63M parameters respectively, compared to only 70K in BLEND.
>
> **Limitations And Societal Impact**
>
> We thank the reviewer for highlighting these potential issues. We would like to emphasize that our paper per se does not study such potential negative applications. The mathematical framework we develop could allow us to better interpret and understand existing GNN models. We believe that better understanding ML models is key to understanding their potential societal implications and preventing their negative societal impacts.  We will reword the Impact Statement accordingly.

---

> > ### Comment · Reviewer_RFfM · 2021-08-24
> > **Thank you for your reply**
> >
> > Thank you for your clarifications. I heavily disagree with your take on the societal impact, even if you do not *study* it, if your method gets better results *in general* and it's mainly applied to socially dubious things, it warrants thinking about mitigation, impact and acknowledgement at least. The very least we can do when developing methods that might be abused is to be ashamed of it. So I urge you to at least mention them in the societal impact section (as well as, on a technical level, mentioning the fact that a study of the general expressiveness of GNN+rewiring methods remains an open work)
> >
> > As per the the other remarks, thank your for the elaboration here and for the ones you will include in the paper. Given that
> >
> > - the run time seems to be "not too bad" in terms of scaling (i.e. similar or better scaling than GAT looking at your experiments, with a larger constant factor)
> > - the results are either the same (modulo noise) or strictly better
> > - my other questions seem to be answered
> >
> > I'll upgrade my score

---

> > > ### Author Response · Authors · 2021-08-24
> > > **Thanks**
> > >
> > > Thanks for increasing the score. We will rephrase the impact statement as per your suggestion.

---

### Official Review · Reviewer_3zke · 2021-07-15

**Rating:** 7
**Confidence:** 4

**Summary:**

The paper proposes a new graph neural network based on a continuous time flow. Each node is given a pre-processed position from an off-line node embedding method, then both the position and a vector of node features (together called Z) are changed during the flow. The flow changes Z by the average difference of Z of adjacent nodes, weighted by a scaled dot product attention. Additionally, a node-wise initial encoder and final decoder is used. The flow is computed via explicit adaptive or fixed integration and the backward pass via the adjoint method. The authors propose various extensions, including having dynamic adjacency via kNN and using hyperbolic positional space. The authors claim their method is motivated as a discrete analogue of a Beltrami flow.

**Limitations And Societal Impact:**

The fact that their theory does not seem to be applicable to the used model, is not honestly mentioned in the limitations. To the contrary, the vagueness of unspecified 'structural assumptions', that are only given in the appendix, makes this theoretical limitation hard to find.

I think the authors underestimate the current use of graph neural networks in industry. They are used widely. As such, some more elaboration on potential negative societal impact of graph neural networks in general could be given.

**Main Review:**

### Strengths:
- The method performs well on smaller data sets with a small number of parameters.
- The method explores some interesting new ideas for graph neural networks.

### Weaknesses:
- I am not quite convinced by the motivation of the proposed method as a discrete analogue of the continuous Beltrami flow. The “structural assumptions on the diffusivity” $a$ seem to not be satisfied scaled dot product attention in BLEND. What is the point of all the theoretical motivation if the actual construction violates the assumptions of the theory?
- Currently, I find it unclear which aspect of the proposed method makes it perform well. Is it the position that is added as a pre-processing step, the continuousness of the flow, the particular integrator, or the particular flow equation (6)? The ablation analysis in the appendix only partially answers this question. Further experiments could include: augmenting GAT with positional encodings; using BLEND with Euler steps; using GAT with continuous integration.
- The method does not seem to get state-of-the-art results on larger sized data sets. The GAT baseline uses more parameters, but would BLEND improve if it used as many? I expect that BLEND has much higher training and inference time than GAT, even with the smaller model, because of the continuous integration. Concrete run-times are not given, so I can’t say for sure.

### Further suggestions for improvement:
- I’d be very interested in the performance of including channel mixing in the flow, referred to as Onsager diffusion, as currently, the fact that the channels only interact via the attention seems limiting. The same holds for time-dependent diffusivity.
- Include (some idea of) the structural assumptions in thm 1 in the main paper.
- Include results on BLEND-kNN on ogb-arxiv or explain why this result is missing.
- Include some actual run-times of BLEND(-kNN) vs other methods.
- In the appendix clarify why (9, suppl mat) is the obvious discrete analogue of (6, suppl mat). I see some notational similarity between (10, suppl mat) and (6, suppl mat), but that looks rather superficial, besides it being one possible generalization of the classic Dirichlet energy. In fact, the paper doesn’t seem to be using this additional generality and only shows the classic example in which $\tilde\psi$ is constant. What does this additional $\psi$ generality add?
- Could the authors clarify the step from (8, suppl mat) to (1)? Where does the time come from?

### Typos:
-	Eqn 4, $x(x,0)$ should be $x(v, 0)$
-	In Table 3, the score for GCN on CiteSeer is bold.
-	The colouring in Table 3 seems incorrect. The BLEND-kNN performs on par with CGNN.
-	The eqn under line 48 in the suppl mat would be clearer if parenthesis would be added to indicate that the partial derivative only applies to Z.
-	Line 153, missing reference


### Conclusion:
- Originality: The work is original. It tries to connect differential geometry to continuous flows in graphs in a way I hadn’t seen before.
- Quality: I have doubts about the correctness, as I question whether the presented theory applies to the proposed model. Additionally, important ablations are missing.
- Clarity: The paper is clearly written.
- Significance: The paper can be significant to all people researching graph neural nets and open exploration into continuous flows on this domain.
- Score: 5, marginally below acceptance threshold. If the authors can convince me the theory does apply to their model, I will increase my score.
- Confidence: 4. I read the paper in detail.


**Time Spent Reviewing:**

8

---

> ### Author Response · Authors · 2021-08-10
> **Initial response to Reviewer 3zke**
>
> We thank the Reviewer for the careful consideration of our work and praising the originality of our approach and novel links to differential geometry. We provide detailed answers which we believe address the main concerns regarding the theoretical part and the ablation experiments.
>
> **Time-dependent diffusivity and channel mixing**
>
> We agree that both time-dependent diffusivity and channel mixing are interesting extensions that we are currently developing. We believe that the Beltrami flow is a natural framework to design such architectures. In particular, weight mixing emerges naturally from forms of attention / diffusivity that satisfy the structural requirements that make the graph Beltrami flow equivalent to the gradient flow of the discrete Polyakov action. One of the key advantages of time-dependent diffusivity as opposed to layer-wise attention is the ability to parametrize it with a small number of parameters determining the temporal trajectory instead of using different per-layer parameters. We believe this is worth a separate paper and leave it for future research.
>
> **Include (some idea of) the structural assumptions in Theorem 1 in the main paper.**
>
> Thanks for this suggestion. We will add a brief discussion about the derivation of the graph Beltrami flow as the minimization of the discrete Polyakov functional and the assumptions on the structure of the PDE.
>
>
> **Include results on BLEND-kNN on ogb-arxiv or explain why this result is missing.**
>
> The “kNN” version has a pre-processing stage where the graph is rewired using the DIGL methodology of Klicpera et al. 2019 (see Section 3.3), which constructs an n-dimensional representation of each node (n-by-n matrix), then sparsifies into a kNN graph. The ogb-arxiv dataset has >150K nodes making this computation run out of memory. Note that this is not a limitation of BLEND but that of DIGL rewiring. Other forms of initial rewiring are possible, but as we chose to include a comparison using DIGL (which is perhaps the most popular version of graph rewiring) exactly like in the Klicpera et al., this result is missing.
>
> **Further experiments could include: augmenting GAT with positional encodings; using BLEND with Euler steps**
>
> Please see response to Reviewer eTTt **Compare with explicit Euler** for an ablation against BLEND with Euler steps and below for an ablation of GAT using positional encodings
>
> |               | Cora           | Citeseer       | Pubmed         | CoauthorCS     | Computers       | Photo           |
> |---------------|----------------|----------------|----------------|----------------|-----------------|-----------------|
> | BLEND w/o POS | 83.6 &pm; 1    | 73.4 &pm; 0.5  | 78.8 &pm; 1.7  | 92.9 &pm; 0.4  | 83.7 &pm; 1.2   | 92.3 &pm; 0.9   |
> | BLEND         | 84.8 &pm; 0.9  | 75.9 &pm; 1.3  | 79.5 &pm; 1.4  | 92.9 &pm; 0.2  | 86.9 &pm; 0.6   | 92.9 &pm; 0.6   |
> | GAT w. POS    | 82.59 &pm; 1.2 | 74.54 &pm; 1.9 | 76.82 &pm; 1.6 | 90.32 &pm; 0.3 | 73.94 &pm; 23.5 | 76.72 &pm; 29.1 |
>
> Note that the high variances of GAT w. POS for Computers and Photo are consistent with those reported without positional encodings in Pitfalls of Graph Neural Network Evaluation and our paper.
>
> **Include some actual run-times of BLEND(-kNN) vs other methods**
>
> We include below the training and inference runtimes of BLEND and BLEND-kNN, compared with corresponding runtimes of GAT. We use a standard GAT implementation (also used in our Results Table) with 2 layers and 8 heads (all other hyperparameters  are tuned). Experiments used a Tesla K80 GPU with 11GB of RAM.
> Training  times
>
> *Training  times (s) for 100 epochs*
>
>
> |                  | Cora  | Citeseer | Pubmed | CoauthorCS | Computers | Photo |
> |------------------|-------|----------|--------|------------|-----------|-------|
> | BLEND (s)     | 26.79 |    27.74 | 168.71 |     183.29 |     230.2 | 69.85 |
> | BLEND_kNN (s) | 44.65 |    83.84 | 387.86 |     216.94 |    157.77 | 73.86 |
> | Av. degree       |   4.1 |      3.5 |    4.5 |        8.9 |      36.7 |  31.8 |
> | Av. degree kNN   |    32 |       64 |   61.1 |       12.4 |        22 |  38.2 |
> | GAT (s)       |  1.68 |     1.76 |   9.85 |      18.96 |     27.12 | 10.65 |
> | BLEND / GAT      | 15.95 |    15.72 |  17.12 |       9.67 |      8.49 |  6.56 |
> | BLEND_kNN / GAT  | 26.59 |    47.51 |  39.36 |      11.44 |      5.82 |  6.93 |
>
> *Inference  times (s)*
>
> || Cora | Citeseer | Pubmed | CoauthorCS | Computers | Photo |
> |------------------|----------------------------|--------------------|--------------------|--------------------|--------------------|--------------------|
> | BLEND (s) | 0.0712 | 0.0878 | 0.2955 | 0.4490 | 0.4603 | 0.2071 |
> | BLEND_kNN (s) | 0.1140 | 0.2458 | 0.9463 | 0.4945 | 0.3029 | 0.2246 |
> | Av. degree | 4.1 | 3.5 | 4.5 | 8.9 | 36.7 | 31.8 |
> | Av. degree kNN | 32 | 64 | 61.1 | 12.4 | 22 | 38.2 |
> | GAT (s) | 0.0031 | 0.0039 | 0.007 | 0.0326 | 0.0165 | 0.0093 |
> | BLEND / GAT | 22.72 | 22.46 | 42.5 | 13.8 | 27.91 | 22.35 |
> | BLEND_kNN / GAT | 36.4 | 62.88 | 136.08 | 15.19 | 18.37 | 24.23 |
>
>
> These results will be added to future versions of the manuscript.
>
> **I think the authors underestimate the current use of graph neural networks in industry... more elaboration on potential negative societal impact of graph neural networks in general could be given.**
>
> While we agree that GNNs have recently become increasingly utilized in industrial applications, and hence could potentially lead to a negative societal impact if used improperly, we would like to emphasize that our paper per se does not study such potential negative applications. The mathematical framework we develop could allow us to better interpret and understand existing GNN models. We believe that better understanding ML models is key to understanding their potential societal implications and preventing their negative societal impacts.  We will reword the Impact Statement accordingly.
>
>
>
> **Could the authors clarify the step from (8, suppl mat) to (1)? Where does the time come from?**
>
> Equation (1) derives from (8) in SM as a gradient flow, meaning that the evolution in time of the signal is given by the negative gradient of the functional. A full derivation can be found e.g. in [40].
>
>
> **In the appendix clarify why (9, suppl mat) is the obvious discrete analogue of (6, suppl mat)... What does this additional generality add?**
>
> In our theoretical analysis, \psi plays the role of discrete counterpart of the Riemannian metric on a manifold, as there is no immediate analogy of pullback on graphs. The additional generality stems from the fact that such \psi can depend on the embedding itself, an exact analogy of the continuous setting where the Riemannian metric is induced via the embedding. This leads to equations with the coupling diffusivity depending on the embedding, which can be interpreted as an attention mechanism.
>
> **I question whether the presented theory applies to the proposed model... If the authors can convince me the theory does apply to their model, I will increase my score.**
>
> We respectfully disagree with this judgement. First, we stress that the PDE we give in Equation 5 is the correct graph version of the graph Beltrami flow (a non-Euclidean generalization of the diffusion equation). The question we attempt to address in the SM is when the graph Beltrami flow can be interpreted as the minimization of “embedding smoothness”, and show that this is possible in some cases. This is thus not a “limitation” but an additional connection that is not completely worked out yet and requires further research.
>
> More specifically:
>
> 1. Similarly to Kimmel et al. (1997), we consider an instance of the Beltrami flow that avoids channel mixing. In the continuous case, it is enough to assume the Euclidean embedding space because the pullback metric produces no channel mixing and yields an evolution equation that has the form of the divergence of the gradient, with the diffusivity acting as a coupling term. The general Beltrami flow has additional connection terms that result in channel mixing (see http://www.math.tau.ac.il/~sochen/Articles/KSM_SC-SP_97.pdf, eq 5). In the graph Beltrami flow, we have the same structure of the equation (divergence of the coupled gradient), with graph gradient and divergence operators. This structure replicates that of GAT and allows us to interpret the coupling term (diffusivity) as the attention and study attention-based GNN models as instances of the graph Beltrami flow.
> 2. In the SM, we consider a general form of the graph Beltrami flow. Since there is no immediate analogy of pullback on graphs, we have to introduce a discrete analogy of the Riemannian metric through the function $\psi$. Avoiding channel mixing therefore requires additional assumptions on $\psi$.
> 3. The continuous Polyakov functional is a generalization of the Dirichlet energy measuring the “smoothness” of a general Riemannian embedding. We are not aware of a discrete version of this functional. The functional we define is analogous to the continuous Polyakov functional in its structure and also can be reduced to the simple graph Dirichlet energy (see e.g. https://dennyzhou.github.io/papers/DR.pdf).
> 4. The general form of the discrete Polyakov functional (SM eq. 9) leads to a gradient flow of the form $f(\psi)(z_i - z_j) + g(\psi) \| z_i - z_j \|^2$ (where $f$ and $g$ are fixed functions, and $\psi$ is a function we can choose). The $g$-term leads to channel mixing. The model we consider has no $g$-term. What we show in the SM is how to choose $\psi$ to make the $g$-term disappear. If the channel mixing $g$-term is ignored, then the structure constraints on $\psi$ are no longer needed. By allowing more flexibility/freedom for the choice of $\psi$ (and hence the attention function), we do not necessarily have the interpretation of a gradient flow of the discrete Polyakov functional.

---

> > ### Comment · Reviewer_3zke · 2021-08-18
> > **Thank you for comments. Questions about theory remain**
> >
> > Thank you for your comments and the additional experimental results.
> >
> > I'm still confused why equation 5 is the obvious discrete version of equation 2, which has a very particular form of the diffusivity. It looks to me more that equation 5 is a very general form, which is a discrete version of equation 2 only if some assumptions on $a$ are satisfied. Can the authors motivate their choice of calling equation 5 *the* discrete Beltrami flow for any choice of $a$, and not an equation that only has some analogy to the continuous Beltrami flow?
> >
> > Thanks

---

> > > ### Author Response · Authors · 2021-08-20
> > > **Discussion about graph Beltrami flow**
> > >
> > > When looking at equation (2) there are two levels of interpretation. One relates to the pde point of view, the other to the diffusivity emerging from the pull-back of the ambient metric.
> > > Namely, assume that we are given a manifold U equipped with some metric g we have no control on, and consider an embedding Z of U into Euclidean space. If we minimize the Polyakov action with respect to the embedding Z only, then we obtain an equation of the form (2). The diffusivity in (2) is going to be the determinant of the metric g. Since there are no constraints on this metric, there is apriori a lot of freedom on the diffusivity coefficient. In fact, from a PDE point of view we simply have an equation of the form div(a*grad), with a > 0.  The discrete counterpart of this equation is then given by 5 which is of the same form. This is at the level of differential equation and motivated the characterization of Graph Beltrami flow.
> > > If we also minimize the embedding Z with respect to the metric g, then such g is no longer a free variable but must be the pull-back of the Euclidean metric via Z. This is where the comparison becomes less obvious because we do not have a discrete analogous of the operation of pull-back of a Riemannian metric on a graph. This is what we try to address in the SM by studying variations of the classical Dirichlet energy on graphs. In this case structural constraints arise because we are trying to reproduce the pull-back constraints.

---

> > > > ### Comment · Reviewer_3zke · 2021-08-24
> > > > **Thanks for clarification**
> > > >
> > > > Thank you for your clarification. I now better understand that there are two points of view and that you claim that (5) is a graph discretisation of the Laplace-Beltrami operator in (2) for an arbitrary metric tensor G - which we'll let arbitrarily depend on the value of the field. I think it'd be good to add to the paper explicitly as a limitation of your theory that the theorem and the SM does not apply to your BLEND flow.
> > > >
> > > > Still, then I don't quite understand why (5) is reasonable. As the determinant of the metric is a scalar field, why is $a$ a function of both $z_i$ and $z_j$? The approach taken by [1, eqn 1.3] does depend on a node-wise weight, not an arbitrary edge-wise weight. Can you comment on the difference between your discretisation of the Laplace-Beltrami operator and the one in [1]?
> > > >
> > > > [1] Burago, Dmitri, Sergei Ivanov, and Yaroslav Kurylev. "A graph discretization of the Laplace–Beltrami operator." Journal of Spectral Theory 4.4 (2015): 675-714.

---

> > > > > ### Author Response · Authors · 2021-08-24
> > > > > **Eqn (5) vs eqn (1.3) in Burago et al.**
> > > > >
> > > > > Thank you for your reply. We will update the manuscript and point out more explicitly in the Limitations section that the theory discussed in Theorem 1 and in the SM does not apply to the BLEND flow implemented.
> > > > >
> > > > > As for the comparison with [1], we note that eqn 1.3 is defined as a discretization of the Laplace-Beltrami operator on some finite $\epsilon$-net. This is a scenario completely different from ours, where we are given an arbitrary graph which in general is not an $\epsilon$-net of a smooth manifold. In fact, such graph is naturally induced with its own discrete differential operators: $\mathrm{div}$ and $\mathrm{grad}$. The point is rather how can we imitate the Beltrami flow by trying to pull back a Riemannian structure from a target ambient space (e.g. the Euclidean space) to the graph and flow the embedding accordingly. This leads to a rather subtle point which we will rigorously investigate in a separate place: how do we define a pullback Riemannian structure on a graph?
> > > > >
> > > > > In this manuscript, we have tried to provide a first ad-hoc answer. As observed above, if we minimize the Polyakov action w.r.t. the embedding only, we obtain a system of equations of the form div(a*grad). This led us to consider equation (5) with the further generalization of allowing a to depend on the embedding as well to both imitate the attention mechanism and be in line with the Perona-Malik flow [Section 4.2, 84].  If finally, we ask ourselves whether the induced equation is the gradient flow of a modified Dirichlet energy, then we obtain Theorem 1 which should be interpreted as in under which structural constraints we are "approximately" pulling back the Euclidean metric on the graph. Once again, this is different from [1], where instead the authors consider which operator approximates the Laplace-Beltrami operator if the given graph is the skeleton of a manifold in a precise sense.

---

> > > > > > ### Comment · Reviewer_3zke · 2021-08-31
> > > > > > **With some rewording, I can agree with the statements in the paper**
> > > > > >
> > > > > > Thank you for your extensive comments to my questions. I think I agree that your motivation of (5) in the above reply is reasonable and I think it would be good if these comments would be also stated in the final version of the paper. In particular that (5) has been generalized to imitate attention.

---

> > > > > > > ### Author Response · Authors · 2021-08-31
> > > > > > > **Rewording**
> > > > > > >
> > > > > > > Thank you - as suggested by the Reviewer, we will reword the statements about equation (5) and related matters according to our comments.

---

> ### Comment · Reviewer_3zke · 2021-08-31
> **Increased score**
>
> I'd like to thank the authors for their additional baselines and theoretical clarification. I've increased my score to 7.

---

### Official Review · Reviewer_eTTt · 2021-07-20

**Rating:** 7
**Confidence:** 4

**Summary:**

I have read the authors' responses, comments from the other reviewers, and the discussion here. The authors have answered my questions satisfactorily. Hence, I am increasing my score to 7.


This paper derives a new class of GNN using ideas from differential geometry. It proposes discretization of the PDE associated with the Beltrami flow for the design of the GNN. Although the explicit discrete approximation of the Beltrami flow on a graph has many interesting similarities to existing attention mechanisms, this framework can be more general with the use of sophisticated differential equation solvers, as presented by the authors. The paper also brings out interesting connections between their model and existing GNN architectures, which in itself, is a nice contribution of this work.



**Limitations And Societal Impact:**

The authors discuss the limitations and societal impacts adequately.

**Main Review:**

The paper is generally well written except for some typos (possibly), for example,

1) In eq. 4, should there be a $(4\pi t)^{-d/2}$ in the RHS? Without it, if $\alpha \rightarrow 0$, eq.4 does not become the convolution in line 74.

2) In the same equation, should $x(\mathbf{x}, 0) \rightarrow x(v, 0)$, and $x(u), x(v) \rightarrow x(u, 0), x(v, 0)$?

3) In eq. 6, should $U(0)= \alpha U$ be $U(0)= U$ based on the description in section 2?

4) In section 4, should $Y = \psi(Z(T))$ be $Y = \zeta(Z(T))$ based on the subsequent discussion?

5) In eq. 8, $Q^{(k)} \neq A^{(k)} - I$, it's actually $Q^{(k)} = \tau A^{(k)} - (\tau -1) I$. However, the definition of $q_{ij}^{(k)}$ is correct.

6) In line 153, \ref to a section is not working.

Other questions/ comments:

1) Is $\alpha$ learned or treated as a hyperparameter? If it is learned, how is the criterion $\alpha \geq 0$ satisfied? If it is a hyperparameter, what is its effect on the classification accuracy?

2) Can the authors compare the explicit scheme in sec. 3.2 with their implementation using numerical solver? This will allow the readers to understand the actual source of the performance improvements.

3) Can the authors discuss the novelty in this work in comparison to existing ODE-based GNN models (reference 69, 96, 100). I appreciate that they conduct experimental comparisons with these related works, but some comments about the methodology as well would be welcome.

4) In the supplementary material, the authors discuss several choices for the diffusivity function modeling but do not report any experimental comparisons.

Overall, my impression is that this work has some novelty in terms of methodology and strong numerical results. But the writing could be significantly improved and the current version is not entirely satisfactory to be published.

**Time Spent Reviewing:**

6 hours

---

> ### Author Response · Authors · 2021-08-10
> **Initial response to Reviewer eTTt**
>
> We thank the reviewer for their careful consideration of our work and praising the novelty of our methodology and strong numerical results. We are particularly grateful for pointing out several typos, which will be corrected. We provide below detailed answers which we believe address the main concerns of the Reviewer.
>
>
> **Is $\alpha$ learned or a hyperparameter**
>
> $\alpha$ is learned implicitly as the features are passed through linear encoder $\phi$ and $\psi$ described in Section 4.
>
> **Compare with explicit Euler**
>
>
> Below is a comparison of BLEND using the adaptive step-size solver Dopri5 vs explicit Euler with step size of 1:
>
> |     Method | Cora          | Citeseer      | Pubmed        | CoathorCS     | Computers     | Photo            |
> |------------|---------------|---------------|---------------|---------------|---------------|------------------|
> | Euler      | 83.5 &pm; 1.7 | 73.3 &pm; 2.2 | 79.2 &pm; 1.3 | 92.5 &pm; 0.1 | 85.0 &pm; 0.7 | 92.3 &pm; 0.3    |
> | Dopri5     | 84.8 &pm; 0.9 | 75.9 &pm; 1.3 | 79.5 &pm; 1.4 | 92.9 &pm; 0.2 | 86.9 &pm; 0.6 | 92.9 &pm; 0.6    |
>
> **Discuss the novelty compared to other Neural ODE benchmarks**
>
> Our approach substantially differs from previous work applying neural ODEs to GNNs in three aspects: 1) goals, 2) generality, and 3) performance.
>
> **Goals**: In GDE, GODE and CGNN, the goal is to adapt neural ordinary differential equations to graphs. In contrast, we consider non-Euclidean partial differential equations and regard GNNs as particular instances of their discretization (both spatial and temporal). We show that many popular GNNs correspond to specific choices of numerical integration schemes for such PDEs and that many such schemes suffer from issues of accuracy and stability. This connection allows us to explore tools from the domain of PDEs to design more principled and better performing GNNs and adopt more sophisticated numerical schemes with better accuracy and stability properties.
>
> **Generality**: First, we show that specific choices of BLEND discretizations lead not only to some of the most popular GNN architectures, but also methods not generally considered to be GNNs (e.g. transformers and Deepsets). Second, as we consider PDEs rather than ODEs, the discretization of the spatial part of the equation amounts to modification of the underlying graph, offering a principled generalization of the various graph rewiring schemes that have become popular in the GNN literature. Third, the use of a non-Euclidean PDE on a joint positional and feature space provides a framework to interpret various positional encoding techniques. We can naturally use spaces with any metric, in particular, extending recent works on hyperbolic graph embeddings. None of the previous techniques explore the link to differential geometry.
>
> **Performance**: BLEND significantly outperforms the aforementioned ODE methods as demonstrated in Tables 2 and 3, dominating GDE, GODE and CGNN across all 10 experiments.
>
> More specifically, GDE, GODE, and CGNN consider Neural ODEs of the canonical form $dx/dt = f(x,t,\theta)$ using a graph neural network as $f$ (GODE), only the message passing component as $f$ (CGNN), or restricting f to be layers of bijective functions on graphs (GDE; since it does not use the adjoint method, strictly speaking, it is a conventional neural network). Furthermore, in CGNN only ODEs with closed form solutions are considered.
>
> In contrast, we regard GNNs as choices of discretizations of non-Euclidean diffusion PDEs and provide a single theoretical framework that incorporates multiple recent GNN architectural features such as attention, graph rewiring, and positional encoding. Specifically by incorporating a learnable diffusivity function, we generalise attention-based GNNs such as GAT. Using a joint positional and feature space, we provide a geometric framework for positional encoding. Allowing the latent node positions to evolve under the PDE, we incorporate the various forms of graph rewiring.
>
> Sanchez-Gonzalez et al. (2019) on the other hand is quite distinct as they are not concerned with GNN design and instead use graph-based ODEs to generate physics simulations.
>
>
> **Ablation with the diffusivity function**
>
> We present below an ablation study that compares the four types of diffusivity function that we considered:
>
> |     Method   | Cora            | Citeseer        | Pubmed          | CoauthorCS      | Computers       | Photo              |
> |--------------|-----------------|-----------------|-----------------|-----------------|-----------------|--------------------|
> | cosine_sim   | 83.8 &pm;   0.5 | 73.5 &pm;   1.1 | 79.0 &pm;   2.3 | 92.8 &pm;   0.1 | 84.8 &pm;   0.7 | 93.2 &pm;   0.4    |
> | exp_kernel   | 83.6 &pm;   1.8 | 75 &pm; 1.3     | 79.9 &pm;   1.4 | 92.8 &pm;   0.2 | 84.8 &pm;   0.5 | 93.2 &pm;   0.5    |
> | pearson      | 83.7 &pm;   1.5 | 75.2 &pm;   1.3 | 80.0 &pm;   1.2 | 92.8 &pm;   0.2 | 84.8 &pm;   0.5 | 93.5 &pm;   0.3    |
> | scaled   dot | 84.8 &pm;   0.9 | 75.9 &pm;   1.3 | 79.5 &pm;   1.4 | 92.9 &pm;   0.2 | 86.9 &pm;   0.6 | 92.9 &pm;   0.6    |
>
> In general scaled dot is the best performing method in 3 out of 6 datasets.

---

> > ### Comment · Reviewer_3zke · 2021-08-18
> > **Time step 1, for how many steps?**
> >
> > Dear authors,
> >
> > You say that in the comparison to explicit Euler, you use time step-size 1. For how long do you then integrate? In other words, how many 'layers' are there? Did you experiment with other time step-sizes? I wouldn't be surprised if that may significantly affect performance.
> >
> > Thanks

---

> > > ### Author Response · Authors · 2021-08-19
> > > **Integration time, layer counts and varying step size**
> > >
> > > Please see a table that includes the final integration times and number of layers for adaptive dopri5 and the explicit Euler method with step sizes between $2^{-2} \to 2^{3}$.
> > >
> > > |        | Step       | Cora                 | Citeseer             | Pubmed               | CoauthorCS           | Computers            | Photo                |
> > > |--------|------------|----------------------|----------------------|----------------------|----------------------|----------------------|----------------------|
> > > |    Time    |       | 18.3                 | 7.9                  | 12.9                 | 3.1                  | 3.2                  | 3.6                  |
> > > | Dopri5 | adaptive     | 16.7                 | 23.9                 | 10.4                 | 8                    | 14                   | 10.4                 |
> > > | Euler  |     0.250  |              74.00   |              32.00   |              52.00   |              13.00   |              13.00   |              15.00   |
> > > |        |     0.500  |              37.00   |              16.00   |              26.00   |                7.00  |                7.00  |                8.00  |
> > > |        |     1.000  |              19.00   |                8.00  |              13.00   |                4.00  |                4.00  |                4.00  |
> > > |        |     2.000  |              10.00   |                4.00  |                7.00  |                2.00  |                2.00  |                2.00  |
> > > |        |     4.000  |                5.00  |                2.00  |                4.00  |                1.00  |                1.00  |                1.00  |
> > > |        |     8.000  |                3.00  |                1.00  |                2.00  |                1.00  |                1.00  |                1.00  |
> > >
> > > The final integration times are the optimal values based on the hyperparameter search reported in the paper. For the dopri5 results, different initialisations lead to different numbers of layers and the results presented are the average over 8 runs.
> > >
> > > Indeed, your intuition is correct and this does have a significant effect on performance, which is given in the table below. For large step sizes performance drops significantly.
> > >
> > > |        | Step | Cora | Citeseer | Pubmed | CoauthorCS | Computers | Photo |
> > > |--------|---------------|-------------|-------------|-------------|-------------|-------------|-------------|
> > > | Time | | 18.3 | 7.9 | 12.9 | 3.1 | 3.2 | 3.6 |
> > > | Dopri5 | adaptive | 83.8&pm;1.3 | 74.3&pm;2.3 | 80&pm;1.1   | 92.8&pm;0.1 | 85.1&pm;0.5 | 92.8&pm;0.4 |
> > > | Euler  | 0.250  | 84.3&pm;0.8 | 73.1&pm;1.8 | 78.8&pm;1.4 | 92.8&pm;0.2 | 84.8&pm;0.6 | 92.9&pm;0.3 |
> > > | | 0.500  | 83.4&pm;1.1 | 74.2&pm;1.6 | 79.8&pm;1.5 | 92.7&pm;0.3 | 85&pm;0.6   | 92.6&pm;0.2 |
> > > |  | 1.000  | 83.5&pm;1.7 | 73.3&pm;2.2 | 79.2&pm;1.3 | 92.5&pm;0.1 | 85&pm;0.8   | 92.3&pm;0.3 |
> > > |  | 2.000  | 84.3&pm;1.1 | 74.9&pm;1.1 | 78.6&pm;1.9 | 91.6&pm;0.1 | 84.8&pm;0.4 | 91.7&pm;0.5 |
> > > |  | 4.000  | 40.9&pm;3.1 | 59.9&pm;3.8 | 72.4&pm;2.5 | 77.1&pm;0.3 | 75.8&pm;1.1 | 82.7&pm;0.8 |
> > > |  | 8.000  | 35.1&pm;2.8 | 23.3&pm;3.2 | 46.8&pm;1.9 | 77.1&pm;0.4 | 75.2&pm;0.8 | 83.2&pm;0.8 |

---

### Author Response · Authors · 2021-08-10
**General remarks**

We would like to thank the Reviewers for carefully considering our work and providing detailed and extensive remarks and suggestions for improving our paper. We are glad that the Reviewers considered the paper in favorable terms: "well written" (eTTt, RFfM), has "novelty in terms of methodology" and "strong numerical results" (eTTt), "original" and "can be significant to all people researching graph neural nets" (3zke), and that "the perspective of graph rewiring through a continuous process is intriguing" (RFfM).

The main questions on which Reviewers asked for for further details are as follows:

**More detailed discussion of BLEND vs graph Neural ODEs:**  see response to Reviewer eTTt.

**Additional ablation studies:** see response to Reviewer eTTt

**Include complexity/runtimes:** see response to Reviewer 3zke

**The applicability of the theoretical analysis in the SM to the model used in the experiments:** see response to Reviewer 3zke

We have provided individual responses to each Reviewer to address these and additional specific questions and concerns. We are also grateful for spotting typos that will be corrected, and will be glad to respond to further questions and provide additional clarifications. We hope in light of these responses that the reviewers will consider increasing their scores.

---

### Decision · Program_Chairs · 2021-09-27

**Decision:**

Accept (Poster)

**Comment:**

The paper proposes a novel class of GNNs based on non-Euclidean diffusion PDEs, i.e., Beltrami flows on 2-manifolds of position and feature embeddings, and can be seen to generalize various existing GNN architectures. The paper is well written and relevant to the NeurIPS community. All reviewers and the AC support acceptance for its contributions, especially due to the novel and interesting ideas that provide a new perspective for graph learning methods as well as the promising empirical results. The authors' rebuttal, including additional experiments, further increased the confidence of reviewers and resolved concerns related to the empirical analysis. When preparing the camera ready version, please incorporate the overall feedback of reviewers into the new revision (e.g., motivation, theoretical discussion of Thm 1 and limitation, ablation).